# TeDS: Joint Learning of Diachronic and Synchronic Perspectives in Quaternion Space for Temporal Knowledge Graph Completion

**Jiujiang Guo** [1 2 3]  **Mankun Zhao** [1 2 3]  **Wenbin Zhang** [1 2 3]  **Tianyi Xu** [1 2 3]  **Linying Xu** [1]  **Jian Yu** [1 2 3]  **Mei Yu** [1 2 3]  **Ruiguo Yu** [1 2 3]

## Abstract

Existing research on temporal knowledge graph completion treats temporal information as supplementary, without simulating various features of facts from a temporal perspective. This work summarizes features of temporalized facts from both diachronic and synchronic perspectives: (1) Diachronicity. Facts often exhibit varying characteristics and trends across different temporal domains; (2) Synchronicity. In specific temporal contexts, various relations between entities influence each other, generating latent semantics. To track above issues, we design a quaternion-based model, TeDS, which divides timestamps into diachronic and synchronic timestamps to support dual temporal perception: (a) Two composite quaternions fusing time and relation information are generated by reorganizing synchronic timestamp and relation quaternions, and Hamilton operator achieves their interaction. (b) Each time point is sequentially mapped to an angle and converted to scalar component of a quaternion using trigonometric functions to build diachronic timestamps. We then rotate relation by using Hamilton operator between it and diachronic timestamp. In this way, TeDS achieves deep integration of relations and time while accommodating different perspectives. Empirically, TeDS significantly outperforms SOTA models on six benchmarks.

---

*Equal contribution [1]College of Intelligence and Computing, Tianjin University, Tianjin 300354, China [2]Tianjin Key Laboratory of Advanced Networking, Tianjin University, Tianjin 300354, China [3]Tianjin Key Laboratory of Cognitive Computing and Application, Tianjin University, Tianjin 300354, China. Correspondence to: Ruiguo Yu <rgyu@tju.edu.cn>.

## 1. Introduction

Knowledge graphs (KGs) are a technology that expresses knowledge in a structured form, which is generally classified into static knowledge graphs (SKGs) and temporal knowledge graphs (TKGs). SKGs and TKGs store facts as triples and quadruples. Existing KGs (e.g., ICEWS (Lautenschlager et al., 2015), NELL, and OpenIE) contain a vast amount of information, and many have been successfully applied in various domains, including community detection (Li et al., 2024), failure KGs for offshore wind power (Ding et al., 2025), and industry-oriented KGs.

With exponential growth of data, KGs consistently encounter issues such as incompleteness, sparsity, and imbalance. To provide sustainable solutions, researchers focus extensively on knowledge graph completion (KGC), developing a variety of robust models. However, most models focus on SKGs, with less attention to TKGs with temporal information (Guo et al., 2024). Unlike SKGs, TKGs align more closely with real-world scenarios due to inclusion of time dimension, making completion process significantly more challenging. Existing temporal knowledge graph completion (TKGC) research regards temporal information as supplementary, failing to observe various features and trajectories that facts can present from a temporal perspective. To uncover characteristics of facts from temporal perspective, we summarize features of temporalized facts from both diachronic and synchronic perspectives: **(a) Diachronicity.** Facts often exhibit varying characteristics and developmental trends across different temporal domains. For example, as shown in Figure 1, in March, *James* is friends with *Emma* and *Sophia*. By May and June, *James* starts dating *Emma*, and their relation progresses smoothly into a romantic one by July. Meanwhile, *James* remains ordinary friends with *Sophia*. **(b) Synchronicity.** In specific temporal contexts, various relations between entities often influence each other, thereby generating latent semantics. For example, as shown in Figure 1, in May and June, relation between *Emma* and *James* noticeably becomes closer, showing signs of ambiguity. Meanwhile, during this period, *James* and *Sophia* only have one call, remaining ordinary friends. The above perspectives are not entirely independent, they often mutually

depend on and serve as foundations for each other. Thus, designing a unified framework to simultaneously consider both diachronicity and synchronicity is an interesting work.

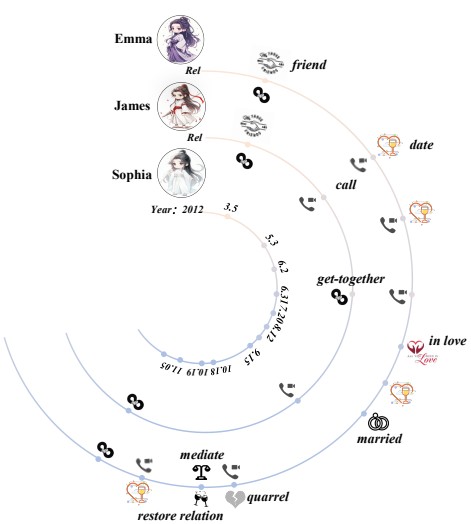

*Figure 1.* A brief illustration reflecting diachronic and synchronic facts.

The primary purpose of our work is to introduce a novel quaternion-based model TeDS. Specifically, TeDS divides timestamp into diachronic timestamp and synchronic timestamp to design dual temporal perception channels for tracking synchronicity and diachronicity, as follows: (a) **Synchronic perception.** We reorganize synchronic timestamp and relation quaternions to construct two composite quaternions that effectively blend time and relation information. We then use Hamilton operator to achieve their interaction, further achieving interaction between relations and time. (b) **Diachronic perception.** We iterate through all timestamps, sequentially mapping each time point to an angle and converting it to the scalar component of a quaternion using trigonometric functions, thereby constructing diachronic timestamps to interpret timestamps continuously. We then rotate the relation using Hamilton operator between it and the diachronic timestamp, enabling the relation to carry diachronic information. In this way, TeDS can achieve deep integration of relations and time while accommodating different temporal perspectives.

- We observe the regularities of facts within a temporal context and summarizes two important temporal perspectives: synchronicity and diachronicity.

- We utilize the representation capability of quaternions to design dual perception channels and integrate them into a unified framework to address the different perspective features of various temporalized facts.

- We empirically demonstrate that TeDS achieves significant improvements over the existing SOTA on different datasets. Furthermore, TeDS underwent detailed strict constraint experiments and withstood the tests.

## 2. Related Work

### 2.1. SKGC models

We categorize SKGC models into five families. **Translation Family.** TransE interprets relations as a translation between head and tail entity to achieve its modeling objectives. Subsequent works in this line, such as TransD (Ji et al., 2015) and TransR. **Complex Family.** Learning KGs in complex space has gained attention. ComplEx (Trouillon et al., 2016) and RotatE (Sun et al., 2019) extend Euclidean space into complex space. Besides, QuatE (Zhang et al., 2019) and DualE (Cao et al., 2021) treat relation as a rotation in quaternion space and dual quaternion space. **Tensor Family.** Tensor decomposition and its properties are effective at deconstructing and capturing logical rules and latent semantics. RESCAL (Nickel et al., 2011) incorporates a bilinear multiplication between embeddings for both head and tail entity, along with a full-rank matrix unique to each relation. TuckER (Balazevic et al., 2019), based on Tucker decomposition, factorizes fact into a core tensor and three member matrices in which each row is embedding of corresponding entity or relation. Recently, BDRI (Yu et al., 2023a) leverages BTD structure and inverse patterns to enhance forward-inverse relation interactions. **Compound Family.** Current research integrates advantages of different geometric spaces into a unified framework to get synergistic gains. CompoundE (Ge et al., 2023) combines compound transformations to align design variants with rich relation patterns. **Deep learning Family.** Some models using neural networks to accomplish SKGC task gets remarkable results. For example, EIGAT (Zhao et al., 2022), HyperGatE (Fang et al., 2025), and RGAI (Shang et al., 2024a). Recently, integrating logical rules and geometric information to improve deep learning model interpretability has become a research focus. MGTCA (Shang et al., 2024b) integrates mixed geometry information into a trainable convolutional attention network to enhance KGC.

### 2.2. TKGC models

We divide TKGC models into five families. **Translation Family.** TTransE (Jiang et al., 2016) extends TransE by incorporating timestamp, where timestamp is represented as translation operation. Several works have been proposed along this line. For instance, HYTE (Dasgupta et al., 2018) extends TransH by representing timestamps as a learned hyperplane. **Complex Family.** TComplEx (Lacroix et al., 2020), TPComplEx (Yang et al., 2024), and Joint-MTComplEx (Zhang et al., 2024b) model facts in complex

space. Next, RotateQVS (Chen et al., 2022) and MTE (Yu et al., 2025) respectively represent temporal fact information as quaternion. **Tensor Family.** TASTER (Wang et al., 2023) learns global and local information through sparse transfer matrices to adapt to TKGs of different scales. TeLM (Xu et al., 2021) uses multivectors to perform 4th-order tensor factorization in TKGs. TBDRI (Yu et al., 2023b), BTDG (Lai et al., 2022), and CEC-BD (Yue et al., 2024), based on BTD, utilize different ways to represent facts, respectively. Recently, MvTuckER (Wang et al., 2024) learns multi-view representations using Tucker decomposition, transforming them into an $n$th-order binary tensor. **Mathematical rules Family.** CDRGN-SDE (Zhang et al., 2024a) leverages the characteristics of stochastic differential equation, integrating time and dimensional data segment by segment. **Deep learning Family.** Some works introduce neural networks into TKGC. For example, TeMP-SA (Wu et al., 2020), xERTE (Han et al., 2021), ODETKGE (Huang et al., 2024), DLGR (Xiao et al., 2024), SANe (Li et al., 2024), MDRQS (Zhu et al., 2025), Neo-TKGC (Qiu et al., 2025), and GLARGCN (Wang et al., 2025). A drawback of above models is that their geometric meaning is unclear.

## 3. Preliminaries

This section formally describes TKGC problem and the relevant theoretical concepts employed in our work.

### 3.1. Problem Definition

**Temporal knowledge graph** $\mathcal{G}$ is composed of quadruples consisting of entities, relations, and timestamps. We formulate denote $\mathcal{E}$ as the set of all entities, $\mathcal{R}$ as set of all relations, and $\mathcal{T}$ as set of all timestamps. A quadruple is represented as $(s, r, o, \tau) \subset \mathcal{E} \times \mathcal{R} \times \mathcal{E} \times \mathcal{T}$, where $s \in \mathcal{E}$ and $o \in \mathcal{E}$ denote head and tail entity respectively, $r \in \mathcal{R}$ denotes the relation between them, and $\tau \in T$ denotes timestamp. We use $\Pi$ and $\Pi^- = \mathcal{E} \times \mathcal{R} \times \mathcal{E} \times \mathcal{T} - \Pi$ to represent the set of observed quadruples and the set of unobserved quadruples, respectively. Timestamp $\tau$ has multiple cases, such as time period $[\tau_b, \tau_e]$, missing beginning time period $[-, \tau_e]$, missing ending time period $[\tau_b, -]$ and time point $\tau$.

### 3.2. Quaternion Background

**Quaternion** (Hamilton, 1844) typically expressed as $Q = a + e\mathbf{i} + f\mathbf{j} + g\mathbf{k}$, where $a, e, f, g$ are real numbers, $\mathbf{i}, \mathbf{j}, \mathbf{k}$ are imaginary units that satisfy Hamilton rule.
**Conjugate:** The conjugate of $Q$ is $\overline{Q} = a - e\mathbf{i} - f\mathbf{j} - g\mathbf{k}$.
**Norm:** The norm of $Q$ is $|Q| = \sqrt{a^2 + e^2 + f^2 + g^2}$.
**Inner product:** The inner product of $Q_1 = a_1 + e_1\mathbf{i} + f_1\mathbf{j} + g_1\mathbf{k}$ and $Q_2 = a_2 + e_2\mathbf{i} + f_2\mathbf{j} + g_2\mathbf{k}$ is obtained by adding the inner product of the corresponding vector component:

$$Q_1 \cdot Q_2 = \langle a_1, a_2 \rangle + \langle e_1, e_2 \rangle + \langle f_1, f_2 \rangle + \langle g_1, g_2 \rangle. \quad (1)$$

**Hamilton operator** does not satisfy the commutative rule, which means that $Q_1 \otimes Q_2 \neq Q_2 \otimes Q_1$. In addition to commutative rule, the associative and distributive rules hold within quaternions. The product of $Q_1 = a_1 + e_1\mathbf{i} + f_1\mathbf{j} + g_1\mathbf{k}$ and $Q_2 = a_2 + e_2\mathbf{i} + f_2\mathbf{j} + g_2\mathbf{k}$ is as follows:

$$\begin{aligned} Q_1 \otimes Q_2 = &(a_1a_2 - e_1e_2 - f_1f_2 - g_1g_2) \\ &+ (a_1e_2 + e_1a_2 + f_1g_2 - g_1f_2)\,\mathbf{i} \\ &+ (a_1f_2 - e_1g_2 + f_1a_2 + g_1e_2)\,\mathbf{j} \\ &+ (a_1g_2 + e_1f_2 - f_1e_2 + g_1a_2)\,\mathbf{k}. \end{aligned} \quad (2)$$

## 4. TeDS for TKGC

Here, we formally introduce TeDS, which embeds TKGs in quaternion space and scores a quadruple using the Hamilton operator to enhance knowledge representation. Specifically, TeDS utilizes dual temporal perception channels as follows: (a) **Synchronic perception.** We reorganize synchronic timestamp and relation quaternions to construct two composite quaternions. We then use Hamilton operator to achieve their deep interaction. (b) **Diachronic perception.** We iterate through all timestamps, sequentially mapping each time point to an angle and converting it to the scalar component of a quaternion using trigonometric functions, thus constructing diachronic timestamps for continuous interpretation. We then rotate the relation using Hamilton operator between it and diachronic timestamp.

**Symbol description.** Suppose that $\mathcal{G}$ consists of N entities, M relations, and T timestamps. We use the quaternion matrix $Q \in \mathbb{H}^{N \times k}$ to denote all entity embeddings, $R \in \mathbb{H}^{M \times k}$ to denote all relation embeddings, $W \in \mathbb{H}^{T \times k}$ to denote all timestamp embeddings, where each row is an embedding vector for a specific entity of dimensionality $k$. Given a quadruple $(s, r, o, \tau)$, head entity $s$, relation $r$, and tail entity $o$ correspond to $Q_s = \{ a_s + e_s\mathbf{i} + f_s\mathbf{j} + g_s\mathbf{k} : a_s, e_s, f_s, g_s \in \mathbb{R}^k \}$, $R_r = \{ a_r + e_r\mathbf{i} + f_r\mathbf{j} + g_r\mathbf{k} : a_r, e_r, f_r, g_r \in \mathbb{R}^k \}$, and $Q_o = \{ a_o + e_o\mathbf{i} + f_o\mathbf{j} + g_o\mathbf{k} : a_o, e_o, f_o, g_o \in \mathbb{R}^k \}$, respectively. Timestamp $\tau$ is divided into diachronic timestamp $d\tau$ and synchronic timestamp $s\tau$, corresponding to $W_{d\tau}$ and $W_{s\tau}$, respectively.

**Implementation details.** TeDS mainly consists of two modules, the overall architecture is as depicted in Figure 2.

**Synchronic perception (SP).** Firstly, to achieve a thorough integration of temporal and relational information, SP performs a composite reorganization of synchronic timestamp quaternion $W_{s\tau} = a_{s\tau} + e_{s\tau}\mathbf{i} + f_{s\tau}\mathbf{j} + g_{s\tau}\mathbf{k}$ and relation quaternion $R_r$. This results in the creation of two quaternions: $Q_{r\tau_{sp}} = a_r + e_r\mathbf{i} + f_r\mathbf{j} + g_{s\tau}\mathbf{k}$ and $Q_{\tau_{sp}r} = a_{s\tau} + e_{s\tau}\mathbf{i} + f_{s\tau}\mathbf{j} + g_r\mathbf{k}$, where $a_{s\tau}, e_{s\tau}, f_{s\tau}, g_{s\tau} \in \mathbb{R}^k$. This process is visualized in Figure 2(a) and (b). Secondly, we normalize $Q_{\tau_{sp}r}$ to the unit quaternion $Q_{\tau_{sp}r}^{\Delta}$ by dividing $Q_{\tau_{sp}r}$ by its norm to eliminate scaling effects:

$$Q_{\tau_{sp}r}^{\Delta} = \frac{Q_{\tau_{sp}r}}{|Q_{\tau_{sp}r}|} = \frac{a_{s\tau} + e_{s\tau}\mathbf{i} + f_{s\tau}\mathbf{j} + g_r\mathbf{k}}{\sqrt{a_{s\tau}^2 + e_{s\tau}^2 + f_{s\tau}^2 + g_r^2}}. \quad (3)$$

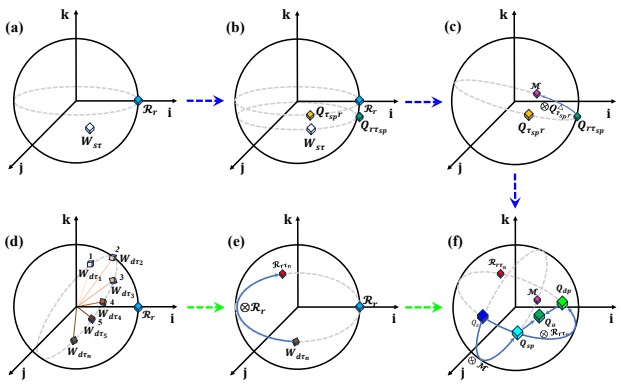

*Figure 2.* Illustrations of TeDS. The blue dashed arrows indicate the process flow of synchronic perceptron; The green dashed arrows indicate the process flow of diachronic perceptron.

Thirdly, as shown in Figure 2(c), TeDS utilizes Hamilton operator to perform rotation operations on $Q_{r\tau_{sp}}$ via $Q^\Delta_{\tau_{sp}r}$, obtaining the representation of synchronic relations:

$$\mathscr{M} = Q_{r\tau_{sp}} \otimes Q^\Delta_{\tau_{sp}r}, \qquad (4)$$

where $\otimes$ is Hamilton operator. Meanwhile, we normalize $\mathscr{M}$ to the unit quaternion $\mathscr{M}^\Delta$. Then, as shown in Figure 2(f), we rotate the head entity $Q_s$ by doing Hamilton operator between it and $\mathscr{M}^\Delta$:

$$Q_{sp} = Q_s \otimes \mathscr{M}^\Delta. \qquad (5)$$

**Diachronic perception (DP).** Firstly, as shown in Figure 2(d), we iterate through all timestamps, sequentially mapping each time point to an angle and converting it to scalar component of a quaternion using continuous trigonometric functions, thereby constructing diachronic timestamps:

$$
\begin{aligned}
W_{d\tau_n} = \cos(\frac{\pi \mathscr{T}_n}{2\mathsf{T}}) + \sin(\frac{\pi \mathscr{T}_n}{2\mathsf{T}})e'\mathbf{i} \\
+ \sin(\frac{\pi \mathscr{T}_n}{2\mathsf{T}})f'\mathbf{j} + \sin(\frac{\pi \mathscr{T}_n}{2\mathsf{T}})g'\mathbf{k},
\end{aligned}
\qquad (6)
$$

where $(e', f', g')$ represents the unit vector of the rotation axis, $e', f', g' \in \mathbb{R}^k$; $\mathscr{T}_n \in \{1, ..., \mathsf{T}\}$ denotes time point. The benefits of above operation are twofold: 1) TeDS can interpret timestamps continuously, "smoothly" capturing time shifts rather than treating each timestamp as discrete. 2) Trigonometric functions provide smooth encoding, enabling TeDS to effectively generalize between frequent and sparse events. Secondly, as shown in Figure 2(e), we use Hamilton operator with the diachronic timestamp $W_{d\tau_n}$ to rotate the relation quaternion $R_r$ into the diachronic relation $R_{r\tau_n}$:

$$R_{r\tau_n} = W_{d\tau_n} \otimes R_r. \qquad (7)$$

Thirdly, we normalize $R_{r\tau_n}$ to the unit quaternion $R^\Delta_{r\tau_n}$. Then, as shown in Figure 2(f), we rotate the head entity $Q_s$

by doing Hamilton operator between it and $R^\Delta_{r\tau_n}$:

$$Q_{dp} = Q_s \otimes R^\Delta_{r\tau_n}. \qquad (8)$$

### 4.1. Scoring function and Loss function

**Scoring function.** As shown in Figure 2(f), we summarize the scoring function of TeDS as follows:

$$\varphi(s, r, o, \tau) = (Q_{sp} + Q_{dp}) \cdot Q_o, \qquad (9)$$

where $\cdot$ is inner product. For the various forms of timestamps, TeDS handles the details as follows: For facts with complete timestamp information $(s, r, o, [\tau_b, \tau_e])$, TeDS splits quadruple into two instances: $(s, r, o, \tau_b)$ and $(s, r, o, \tau_e)$. The score is computed as the average of the two:

$$\varphi(s, r, o, [\tau_b, \tau_e]) = \varphi(s, r, o, \tau_b) + \varphi(s, r, o, \tau_e). \quad (10)$$

For facts with missing timestamp information, i.e., either $(s, r, o, [\tau_b, -])$ or $(s, r, o, [-, \tau_e])$, the score is taken directly from known timestamp:

$$
\begin{aligned}
\varphi(s, r, o, [\tau_b, -]) = \varphi(s, r, o, \tau_b), \\
\varphi(s, r, o, [-, \tau_e]) = \varphi(s, r, o, \tau_e).
\end{aligned}
\qquad (11)
$$

**Optimization.** TKGs can be regarded as fourth-order tensors, so we can use tensor kernel norm regularization to enhance TeDS. Thus, we propose following regularization:

$$\mathfrak{L}_a = \|Q_s\|_4^4 + \|R_r\|_4^4 + \|W_\tau\|_4^4 + \|Q_o\|_4^4, \qquad (12)$$

where $\| \cdot \|_4^4$ denotes nuclear 4-norm. Furthermore, for time constraints, temporal regularizer are commonly used to smooth the representation of adjacent timestamps. Thus, we use the following linear temporal regularizer:

$$\mathfrak{L}_b = \sum_{i=1}^{N} \|W_{\tau_{i+1}-W_{\tau_i}-W_b}\|_4^4, \qquad (13)$$

where $W_b$ is a linear time-constrained bias, initialized randomly and can be learned during training. The linear time constraint can ensure that timestamps of adjacent times are close to each other, and timestamps of distant times are significantly different, ensuring the smoothness of time. And deviation term denotes sudden change of adjacent times.

**Loss function.** Following (Trouillon et al., 2016), we regard TKGC as a multi-classification task, use the cross-entropy loss function:

$$
\begin{aligned}
L = \sum_{(s,r,o,\tau)\in\Pi\cup\Pi'} (-log\frac{exp\left(\varphi(s,r,o,\tau)\right)}{\sum_{s'\in\Pi'} exp\left(\varphi(s',r,o,\tau)\right)} \\
- log\frac{exp\left(\varphi(s,r,o,\tau)\right)}{\sum_{o'\in\Pi'} exp\left(\varphi(s,r,o',\tau)\right)}) \\
+ \lambda_a\mathfrak{L}_a + \lambda_b\mathfrak{L}_b,
\end{aligned}
\qquad (14)
$$

where $\Pi'$ is sampled from the set of unobserved quadruples $\Pi^-$. $\lambda_a$ is the regularization weight and $\lambda_b$ is the temporal regularization weight.

# 5. Experiments and Results

## 5.1. Datasets

We select six common TKGC benchmarks, detailed in Table 1. **ICEWS14**, **ICEWS05-15** and **ICEWS18** are subsets of Integrated Crisis Early Warning System (ICEWS). ICEWS acquires and processes millions of data from various sources to aid in monitoring and responding to global events (e.g., (*Japan*, *Accuse*, *Korea*, *2014-3-3*)). ICEWS14, ICEWS05-15 and ICEWS18 separately collect events from *2014-1-1* to *2014-12-31*, *2005-1-1* to *2015-12-31*, and *2018-1-1* to *2018-12-31*. **YAGO11k** and **Wikidata12k** (Dasgupta et al., 2018) are subsets of YAGO3 and Wikidata, respectively. YAGO3 (Mahdisoltani et al., 2015) and Wikidata (Erxleben et al., 2014) are two TKGs where time annotations are represented in various forms, i.e., time points like (*Allan*, *BornIn*, *Launceston*, [*2005-1-3*, *2005-1-3*]), beginning or end time like (*Biden*, *Presidency*, *POTUS*, [*2021-1-2*, *####-##-##*]), and time intervals like (*Varick*, *Mayor*, *New York*, [*789-3-2*, *801-3-2*]). **GDELT** (Global Database of Events, Language, and Tone) (Trivedi et al., 2017) is a global event database that offers a substantial collection of worldwide facts from *2015-4-1* to *2016-3-31*. It encompasses various events, primarily focusing on political, social, economic, and environmental occurrences.

## 5.2. Experimental Setup

We implement TeDS by running PyTorch. We adopt the Adagrad optimizer to train TeDS. The batch size is fixed as 1000. The regularization weights $\lambda_a$ and $\lambda_b$ are tuned in a range of $\{0, ..., 0.0025, 0.005, 0.0075, 0.01, ..., 0.1\}$. We find the following combinations of regularization weights $\lambda_a$ and $\lambda_b$ to give the best results: (0.0075, 0.01) for ICEWS14; (0.0025, 0.1) for ICEWS05-15; (0.1, 0.01) for ICEWS18; (0.025, 0.001) for YAGO11k; (0.025, 0.0025) for Wikidata12k; (0.00001, 0.01) for GDELT. The optimal embedding dimension is $k = 100$ in all benchmarks.

*Table 1.* Statistics for the various experimental datasets.

|  | ICEWS14 | ICEWS05-15 | ICEWS18 |
|---|---|---|---|
| **Entities** | 6,869 | 10,488 | 23,033 |
| **Relations** | 230 | 251 | 256 |
| **Facts** | 90,730 | 461,329 | 468,558 |
| **Period(year)** | 2014 | 2005-2015 | 2018 |

|  | YAGO11k | Wikidata12k | GDELT |
|---|---|---|---|
| **Entities** | 10,623 | 12,544 | 500 |
| **Relations** | 10 | 24 | 20 |
| **Facts** | 20,507 | 40,621 | 3,419,607 |
| **Period(year)** | 1513-2017 | 1526-2020 | 2015-2016 |

## 5.3. Evaluation Protocol and Baselines

We made the following preparations to evaluate TeDS: 1) **Adopting commonly metrics to evaluate models more accurately**. **Mean reciprocal rank (MRR)** calculates the average score of the reciprocal ranks of the relevant KGs for a given query, where a higher MRR means better performance. **Hit@n** represents the percentage of the top n, where $n \in \{1, 3, 10\}$, where a higher Hit@n indicates better performance. 2) **Selecting a series of state-of-the-art (SOTA) baeslines.** We adopt TTransE (2016), TA-DistMult (2018), TeMP-SA (2020), CyGNet (2021), RE-GCN (2019), TNTComplEx (2020), ChronoR (2021), TeLM (2021), RotateQVS (2022), BTDG (2022), TBDRI (2023), CEC-BD (2024), TPComplEx (2024), Joint-MTComplEx (2024), CDRGN-SDE (2024), MvTuckER (2024), DLGR (2024), ODETKGE (2024), SANe (2024), MTE (2025), MDRQS (2025), Neo-TKGC (2025), and GLARGCN (2025).

## 5.4. Main Results

Table 2 and 3 list results of TeDS and all baseline models on all datasets, where the optimal performance is **bolded**. Overall, TeDS outperforms all baseline models across datasets. Compared with CEC-BD (tensor family), TeDS increases by 27.4 MRR points on ICEWS14, 19.0 MRR points on ICEWS05-15, 17.3 MRR points on ICEWS18, 56.1 MRR points on YAGO11k, 52.2 MRR points on Wikidata12k, and 41.9 MRR points on GDELT. Compared with TPComplEx (complex family), TeDS increases by 0.9 MRR points on ICEWS14; 2.8 MRR points on ICEWS05-15; 30.8 MRR points on GDELT. Compared with SANe (deep learning family), TeDS increases by 26.9 MRR points on ICEWS14, 19.0 MRR points on ICEWS05-15, 52.3 MRR points on YAGO11k, 52.2 MRR points on Wikidata12k.

ICEWS14, ICEWS18 and ICEWS05-15 all belong to ICEWS and thus share similar data types. Compared to its performance on ICEWS14, TeDS shows stronger performance on ICEWS05-15 and ICEWS18. Compared to ICEWS14, ICEWS05-15 covers over 10 times the time span, making its data more reliant on temporal information. This supports TeDS's capability to manage long time series. Given the significant improvement of TeDS on ICEWS18 compared to ICEWS14 and ICEWS05-15, we visualize distribution of facts over time in ICEWS18 and ICEWS, as shown in Figure 3. We find that, at same temporal density, the number of facts in ICEWS18 is approximately 10 times that of ICEWS05-15. This significantly increases the information within the same temporal context, making TKGC more difficult. TeDS addresses this with SP, achieving top performance. Besides, compared to ICEWS, we find that TeDS demonstrates the greatest advantage over SOTA models on YAGO11k and Wikidata12k. To further explore the performance advantages of TeDS, we visualize distribution

*Table 2.* Experimental results on ICEWS14, ICEWS05-15, and ICEWS18. All results are taken from the original papers. Dashes: results are not reported in the responding literature. The best results among all models are written bold.

| | ICEWS14 | | | | ICEWS05-15 | | | | ICEWS18 | | | |
| | MRR | H@1 | H@3 | H@10 | MRR | H@1 | H@3 | H@10 | MRR | H@1 | H@3 | H@10 |
|---|---|---|---|---|---|---|---|---|---|---|---|---|
| TTransE | 25.5 | 7.4 | - | 60.1 | 27.1 | 8.4 | - | 61.6 | - | - | - | - |
| TNTComplEx | 60.7 | 51.9 | 65.9 | 77.2 | 66.6 | 58.3 | 71.8 | 81.7 | - | - | - | - |
| TeMP-SA | 60.7 | 48.4 | 68.4 | 84.0 | 68.0 | 55.3 | 76.9 | 91.3 | - | - | - | - |
| TeLM | 62.5 | 54.5 | 67.3 | 77.4 | 67.8 | 59.9 | 72.8 | 82.3 | - | - | - | - |
| RE-NET | 35.8 | 26.0 | 40.1 | 54.9 | 36.9 | 26.2 | 41.9 | 57.6 | 26.2 | 16.4 | 29.9 | 44.4 |
| RE-GCN | 62.5 | 54.5 | 67.3 | 77.4 | 67.8 | 59.9 | 72.8 | 82.3 | 27.5 | 17.8 | 31.2 | 46.6 |
| RotateQVS | 59.1 | 50.7 | 64.2 | 75.4 | 63.3 | 52.9 | 70.9 | 81.3 | - | - | - | - |
| TBDRI | 65.2 | 55.2 | 69.7 | 78.5 | 70.9 | 64.6 | 75.7 | 82.1 | - | - | - | - |
| BTDG | 60.1 | 51.6 | 65.6 | 75.3 | 62.7 | 53.4 | 68.7 | 79.8 | - | - | - | - |
| TPComplEx | 89.8 | 86.5 | **92.5** | **95.4** | 84.5 | 79.4 | 88.2 | **93.4** | - | - | - | - |
| ODETKGE | 31.1 | 18.4 | 30.2 | 48.2 | 44.0 | 35.4 | 49.4 | 65.8 | 31.5 | 21.0 | 34.3 | **49.3** |
| MvTuckER | 65.4 | 57.7 | 69.9 | 79.7 | 69.8 | 61.8 | 74.7 | 84.1 | - | - | - | - |
| DLGR | 46.7 | 36.6 | 51.6 | - | - | - | - | - | 35.4 | 25.1 | 40.0 | - |
| CDRGN-SDE | 49.2 | 36.5 | 48.4 | 66.9 | 52.3 | 39.8 | 62.5 | 75.4 | - | - | - | - |
| Joint-MTComplEx | 63.6 | 55.6 | 68.1 | 78.1 | 68.3 | 60.1 | 73.6 | 83.2 | - | - | - | - |
| CEC-BD | 63.3 | 55.4 | 68.0 | 77.7 | 68.1 | 60.2 | 73.0 | 82.5 | 28.5 | 18.8 | 32.3 | 47.7 |
| SANe | 63.8 | 55.8 | 68.8 | 78.2 | 68.3 | 60.5 | 73.4 | 82.3 | - | - | - | - |
| MTE | 73.8 | 62.5 | 77.5 | 86.8 | 80.3 | 71.2 | 85.4 | 93.2 | - | - | - | - |
| MDRQS | 62.5 | 54.4 | 67.3 | 77.5 | 67.0 | 59.1 | 71.6 | 81.5 | - | - | - | - |
| Neo-TKGC | 64.2 | 53.7 | 70.9 | 83.6 | 72.2 | 61.7 | 79.4 | 90.8 | - | - | - | - |
| GLARGCN | 67.6 | 62.1 | 71.0 | 77.6 | 77.5 | 72.7 | 80.8 | 85.9 | - | - | - | - |
| TeDS (ours) | **90.7** | **90.6** | 90.8 | 91.0 | **87.3** | **85.8** | **88.3** | 90.4 | **45.8** | **45.6** | **46.9** | 48.2 |

*Table 3.* Experimental results on YAGO11k and Wikidata12k. All results are taken from the original papers. Dashes: results are not reported in the responding literature. The best results among all models are written bold.

| | YAGO11k | | | | Wikidata12k | | | | GDELT | | | |
| | MRR | H@1 | H@3 | H@10 | MRR | H@1 | H@3 | H@10 | MRR | H@1 | H@3 | H@10 |
|---|---|---|---|---|---|---|---|---|---|---|---|---|
| TTransE | 10.8 | 2.0 | 15.0 | 25.1 | 17.2 | 9.6 | 18.4 | 32.9 | 11.5 | 0.0 | 16.0 | 31.8 |
| HyTE | 13.6 | 3.3 | - | 29.8 | 25.3 | 14.7 | - | 48.3 | 11.8 | 0.0 | 16.5 | 32.6 |
| TA-DistMult | 15.5 | 9.8 | - | 26.7 | 23.0 | 13.0 | - | 46.1 | 20.6 | 12.4 | 21.9 | 36.5 |
| TComplEx | 18.5 | 12.7 | 18.3 | 30.7 | 33.1 | 23.3 | 35.7 | 53.9 | 29.8 | 21.3 | 32.3 | 46.4 |
| TeLM | 19.1 | 12.9 | 19.4 | 32.1 | 33.2 | 23.1 | 36.0 | 54.2 | - | - | - | - |
| RotateQVS | 18.9 | 12.4 | 19.9 | 32.3 | - | - | - | - | 27.0 | 17.5 | 29.3 | 45.8 |
| CEC-BD | 21.2 | 15.4 | 21.5 | 33.9 | 33.9 | 24.1 | 36.9 | 54.3 | 29.6 | 20.1 | 33.4 | 46.5 |
| TPComplEx | - | - | - | - | - | - | - | - | 40.7 | 32.9 | 43.1 | 55.9 |
| CDRGN-SDE | - | - | - | - | - | - | - | - | 22.0 | 12.4 | 20.9 | 34.5 |
| MvTuckER | - | - | - | - | - | - | - | - | 54.9 | 47.7 | 58.5 | 68.2 |
| Joint-MTComplEx | 22.2 | 15.8 | 22.9 | 35.6 | 36.0 | 27.0 | 39.3 | 54.1 | - | - | - | - |
| SANe | 25.0 | 18.0 | 26.6 | 40.1 | 43.2 | 33.1 | 48.3 | 64.0 | 30.1 | 21.2 | 32.6 | 47.6 |
| MTE | - | - | - | - | - | - | - | - | 39.1 | 32.2 | 40.9 | 52.2 |
| MDRQS | 27.1 | 19.9 | 29.6 | 40.4 | - | - | - | - | 42.4 | 35.0 | 45.1 | 56.3 |
| Neo-TKGC | - | - | - | - | - | - | - | - | 33.7 | 27.0 | 37.0 | 51.3 |
| GLARGCN | - | - | - | - | 37.3 | 33.8 | 38.5 | 42.3 | 57.1 | 48.4 | 61.3 | 73.0 |
| TeDS (ours) | **77.3** | **76.9** | **78.1** | **78.4** | **95.4** | **95.5** | **95.7** | **95.8** | **71.5** | **69.9** | **72.2** | **74.4** |

of facts in Wikidata12k over time, as shown in Figure 3. We find that facts in Wikidata12k span a long temporal range and exhibit a pronounced long-tail distribution over time, which limits the performance of existing models. TeDS leverages the smooth embeddings provided by DP to effectively handle both short and long temporal spans, enabling it to capture diverse temporal patterns and achieve performance improvements. For a detailed analysis, please refer to ANALYSIS.

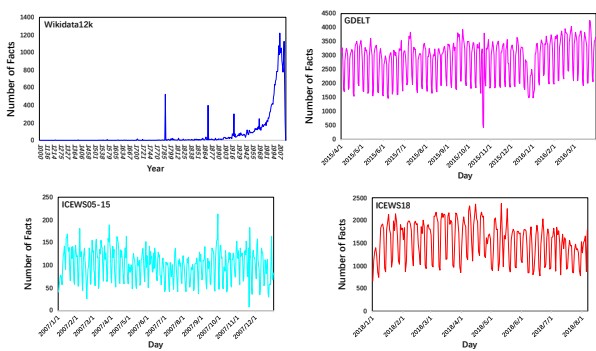

*Figure 3.* The distribution of facts over time in Wikidata12k, GDELT and ICEWS.

# 6. ANALYSIS

We perform ablation experiments on three benchmarks to understand how each perceptron in TeDS handles different types of facts and their benefits. Specifically, we remove the SP in TeDS, reducing it to DP. Next, we remove the DP, reducing TeDS to SP. Besides, we degrade SP into a regular quaternion model named HTM, meaning that the mixing of relations and time is no longer performed.

## 6.1. Ablation study on SP and DP

Table 4 lists specific results of SP and DP on ICEWS14, Wikidata12k, and YAGO11k. SP and DP achieve competitive results compared to SOTA models across all datasets. To further explore mechanisms of different perceptrons, we first observe the distribution of same relation between entities over a long time span, which helps to understand DP's performance. Thus, we extract relation *Consult* between *Barack Obama* and *Benjamin Netanyahu* during 2014 (i.e., (*Barack Obama*, *Consult*, *Benjamin Netanyahu*, *2014-\*-\**)) and visualize the parameters of temporal relation, as shown in Figure 4. Compared to SP and HTM, TeDS enables aggregation of *Consult* within each specific month while ensuring that *Consult* from adjacent months remains close together rather than being randomly distributed. Compared to SP and HTM, TeDS is better at aggregating *Consult* within same month. Besides, TeDS ensures that *Consult* from adjacent months is kept close together rather than being randomly distributed. More specifically, compared to HTM, SP is better at aggregating *Consult* within same month. Unlike TeDS, SP is unable to tightly group *Consult* from adjacent months, although it can distinguish *Consult* from different months to some extent. Meanwhile, compared to existing baseline model TeLM, TeDS is able to aggregate data from different months as much as possible while retaining the ability to classify *Consult* relations across months, thereby clearly delineating the boundaries between adjacent months. It is undeniable that, compared to SP and HTM, TeLM

exhibits a slight advantage in classifying *Consult* across months, which may be attributed to the assistance of temporal embedding regularization. The above experiments and analysis further demonstrate that TeDS is better at modeling the diachronicity of facts.

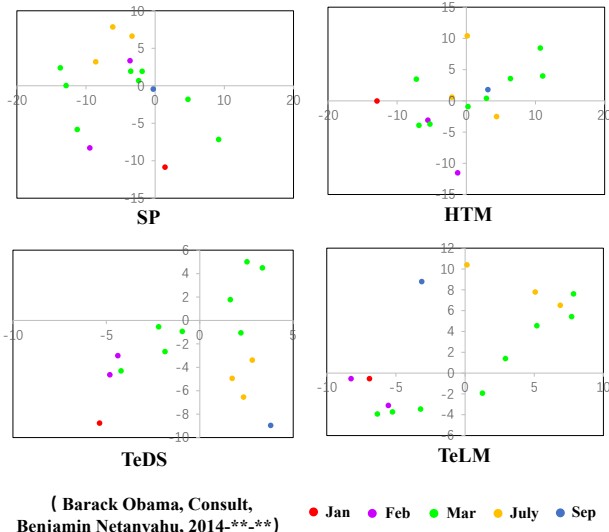

*Figure 4.* Visualizations of temporal relation embeddings learned from different months on ICEWS14, where different colored dots represent the *Consult* in different months.

Next, we observe distribution of various relations between entities within a specific temporal context to better understand SP's performance. Thus, we extract various relations between *Barack Obama* and *Benjamin Netanyahu* from *2014-1* to *2014-6* and visualize the parameters of temporal relation, as shown in Figure 5. Compared to HTM and TeLM, TeDS and SP are better at distinguishing different relations. Moreover, we find that TeDS outperforms SP in distinguishing identical relations, rather than clustering them entirely together. In fact, even within the same month, identical relations can exhibit varied trends depending on the context. Thus, we believe it is crucial to cluster different relations while preserving the unique characteristics of identical relations in specific contexts.

We find that SP outperforms DP across various datasets. To explain this phenomenon, we visualize relation *educated at* among all entities using Wikidata12k. As shown in Figure 6, most facts are concentrated within a specific temporal context (1850 to 2000). Clearly, this aligns with the argument that SP is better at capturing the synchronicity of facts.

*Table 4.* Experimental results of ablation study. The best results among all models are written bold.

| | ICEWS14 | | | | Wikidata12k | | | | YAGO11k | | | |
| --- | --- | --- | --- | --- | --- | --- | --- | --- | --- | --- | --- | --- |
| | MRR | H@1 | H@3 | H@10 | MRR | H@1 | H@3 | H@10 | MRR | H@1 | H@3 | H@10 |
| HTM | 56.4 | 47.8 | 58.3 | 67.5 | 30.2 | 21.4 | 33.0 | 44.7 | 28.8 | 17.8 | 30.1 | 44.3 |
| SP | 79.6 | 76.8 | 80.3 | 83.7 | 67.9 | 59.9 | 72.6 | 83.6 | 48.0 | 38.8 | 53.7 | 64.8 |
| DP | 71.1 | 68.4 | 72.1 | 74.6 | 62.8 | 55.3 | 64.6 | 76.3 | 40.1 | 31.5 | 43.6 | 53.5 |
| TeDS | **90.7** | **90.6** | **90.8** | **91.0** | **95.4** | **95.5** | **95.7** | **95.8** | **77.3** | **76.9** | **78.1** | **78.4** |

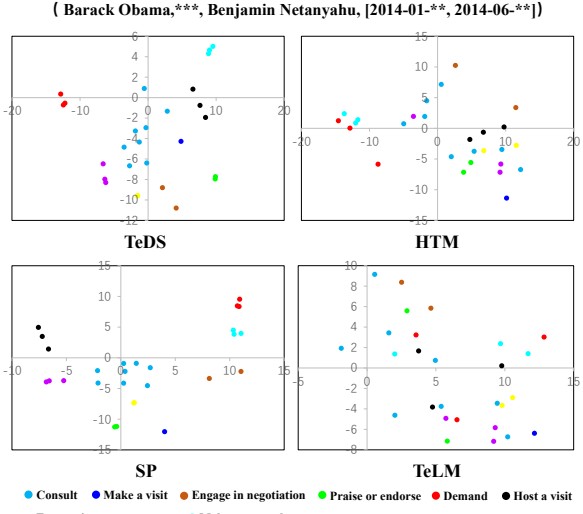

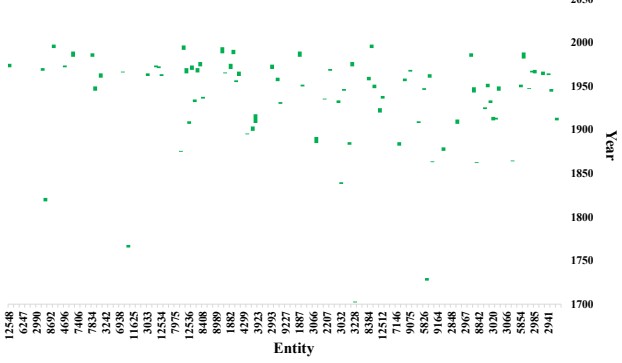

*Figure 6.* The visualization of the relation *educated at* between entities comes from Wikidata12k, where vertical axis represents time, the horizontal axis represents entity, and the length of green rectangles indicates the duration of relation.

*Figure 5.* Visualizations of the temporal relation embeddings learned within specific temporal contexts from ICEWS14, where different colored dots represent different relations.

## 6.2. Strong constraints analysis of quadruples

We extract constrained quadruples from ICEWS14 and ICEWS05-15 to simulate synchronic and diachronic scenarios, as shown in Figure 7. Due to strong constraints of quadruples, we conduct tests using a validation set. For synchronicity testing, we select facts involving *Barack Obama*, *Angela Merkel*, and *Russia* from ICEWS14 between *2014-01-01* and *2014-01-15*. Empirically, TeDS gets most significant results. Moreover, SP consistently outperforms DP and HTM, further validating that SP excels in handling synchronic scenarios. For diachronicity testing, we select facts involving *Barack Obama*, *Angela Merkel*, *Iran*, *Benjamin*, and *Japan* from ICEWS14 and ICEWS05-15 across all time spans. Empirically, TeDS gets most significant performance. Besides, DP consistently outperforms SP and HTM, further validating that DP excels in handling diachronic scenarios. Finally, SP and DP consistently outperform HTM, indicating that SP and DP have a positive impact on both diachronic and synchronic scenarios. This aligns with our motivation: the two perspectives are not entirely independent but often mutually depend on and serve as foundations for each other.

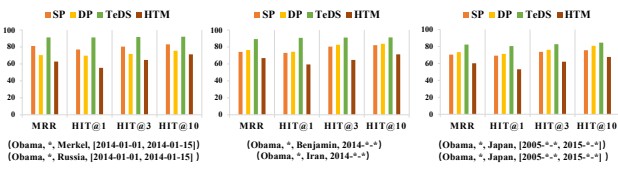

*Figure 7.* Evaluation results on different quadruples in test set. Where [*2014-1-1, 2014-1-15*] represents the years *2014-1-1* to *2014-1-15*; [*2014-\*-\**] stands for the entire year of 2014; [*2005-\*-\*, 2015-\*-\**] represents the years 2005 to 2015; other * represents the rest elements of quadruples in test set that satisfy the current constraints.

## 6.3. Analysis of the robustness and portability of TeDS

We randomly remove 10% and 20% of training set of ICEWS14 (named ICEWS14 10%SPARSE and ICEWS14 20%SPARSE) to assess robustness of TeDS under sparse data conditions, as shown in Figure 8(a) and (b). Empirically, TeDS remains highly competitive compared to HTM on ICEWS14 10% SPARSE and ICEWS14 20% SPARSE, which validates robustness of TeDS. To assess the porta-

bility of TeDS framework, we apply its overall concept to complex embeddings and refer to this extension as CDS. Furthermore, we simplify CDS into a standard complex-valued model named CTM. Besides, we extend the overall concept of TeDS to dual quaternion space, resulting in a variant named DTeDS. The specific results are shown in Figure 8(c) and (d). Experimental results show that TeDS, CDS, and DTeDS outperform HTM and CTM in terms of performance, validating the applicability of our framework. Meanwhile, quaternion model surpasses complex model further confirming superiority of Hamilton operator over Hermitian operator. Finally, DTeDS achieves a slight performance improvement, which we attribute to dual quaternion's greater sensitivity to multi-relational patterns and its richer representational capacity. This will also be a key focus of our future research. The above experiments and analysis further support the portability of TeDS framework.

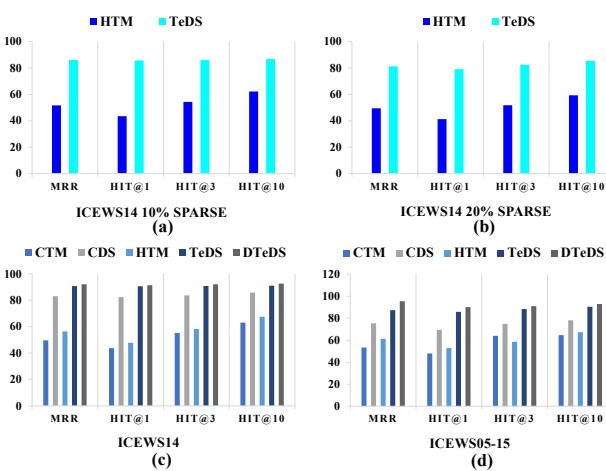

Figure 8. Analysis of the robustness and portability of TeDS

### 6.4. Influence of embedding dimensionality

Figure 9 illustrates MRR performance of TeLM, EHPR, and TeDS on the ICEWS14 dataset across different embedding dimensions $k = \{20, 50, 75, 100, 200, 500, 1000, 2000\}$. Compared to TeLM and EHPR, TeDS consistently achieves significantly better performance at all embedding dimensions and reaches its optimal performance at $k = 100$. In contrast, TeLM and EHPR require higher embedding dimensions to attain their best results. This demonstrates that TeDS not only maintains high accuracy with greater parameter efficiency but also offers more substantial computational advantages as the scale of KGs increases.

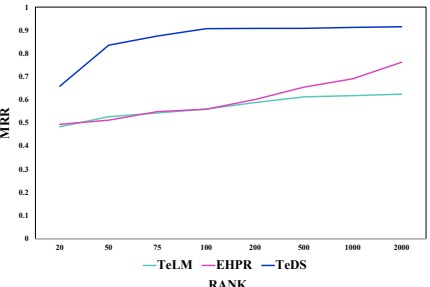

Figure 9. Results of TeDS with different embedding dimensions on ICEWS14.

### 6.5. Training Time and Standard Deviation

We use NVIDIA GeForce RTX 3090 to reproduce the training time for the optimal performance of TComplEx, TeLM, CDS and TeDS on YAGO11k. The specific results are shown in Figure 10(a). Specifically, compared to the SOTA deep learning model (SANe, which takes about 16 hours on YAGO11k), our computation is efficient and controllable. Finally, Figure 10(b) shows the standard deviations for the MRR computed over 5 runs of TeDS on all datasets, demonstrating the stability of TeDS results.

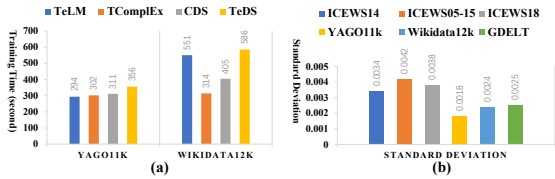

Figure 10. Efficiency and stability evaluation of TeDS.

### 7. Conclusion

This work examines how facts behave over time, highlighting two perspectives: synchronicity and diachronicity. We then use quaternion representation to design two perceptions, integrating them into a unified framework to handle different temporal perspectives. Visualization techniques are employed to test and confirm TeDS's effectiveness in various temporal scenarios. Overall, the analysis and summarization of complex real-world patterns, including the periodicity of facts, remain challenging research problems. In the future, we will continue to focus on various complex scenarios involving temporal facts, such as periodic patterns and complex forms of temporal expressions, and aim to develop more interpretable and context-adaptive models to achieve higher-quality knowledge representations.

## Acknowledgements

This work is supported by the National Key Research and Development Program of China (No.2024YFE0115700) and National Natural Science Foundation of China (No.62402335)

## Impact Statement

This paper presents work whose goal is to advance the field of Knowledge Graph. There are many potential societal consequences of our work, none which we feel must be specifically highlighted here.

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

# B. Appendix

## B.1. HTM for TKGC

This section primarily introduces the regular quaternion model, HTM, which we proposed in ANALYSIS. For HTM, the head entity $s$ and the tail entity $o$ correspond to $Q_s = \{\ a_s + e_s\mathbf{i} + f_s\mathbf{j} + g_s\mathbf{k} : a_s, e_s, f_s, g_s \in \mathbb{R}^k\ \}$ and $Q_o = \{\ a_o + e_o\mathbf{i} + f_o\mathbf{j} + g_o\mathbf{k} : a_o, e_o, f_o, g_o \in \mathbb{R}^k\ \}$, respectively, while the relation $r$ and timestamp $\tau$ correspond to $R_r = \{\ a_r + e_r\mathbf{i} + f_r\mathbf{j} + g_r\mathbf{k} : a_r, e_r, f_r, g_r \in \mathbb{R}^k\ \}$ and $W_\tau = \{\ a_\tau + e_\tau\mathbf{i} + f_\tau\mathbf{j} + g_\tau\mathbf{k} : a_\tau, e_\tau, f_\tau, g_\tau \in \mathbb{R}^k\ \}$, respectively. The specifics are as follows:

We first normalize timestamp $W_\tau$ to the unit quaternion $W_\tau^\Delta$ by dividing $W_\tau$ by its norm to eliminate scaling effects:

$$W_\tau^\Delta(a', e', f', g') = \frac{W_\tau}{|W_\tau|} = \frac{a_\tau + e_\tau\mathbf{i} + f_\tau\mathbf{j} + g_\tau\mathbf{k}}{\sqrt{a_\tau^2 + e_\tau^2 + f_\tau^2 + g_\tau^2}}. \tag{15}$$

Secondly, we rotate relation $R_r$ by doing Hamilton product between it and $W_\tau^\Delta$ to get a representation of temporal relation:

$$R_{r\tau} = R_r \otimes W_\tau^\Delta = a_{r\tau} + e_{r\tau}\mathbf{i} + f_{r\tau}\mathbf{j} + g_{r\tau}\mathbf{k}, \tag{16}$$

where $\otimes$ is Hamilton product. Next, we normalize $R_{r\tau}$ to the unit quaternion $R_{r\tau}^\Delta$:

$$R_{r\tau}^\Delta(a'', e'', f'', g'') = \frac{R_{r\tau}}{|R_{r\tau}|} = \frac{a_{r\tau} + e_{r\tau}\mathbf{i} + f_{r\tau}\mathbf{j} + g_{r\tau}\mathbf{k}}{\sqrt{a_{r\tau}^2 + e_{r\tau}^2 + f_{r\tau}^2 + g_{r\tau}^2}}. \tag{17}$$

We then define an intermediate variable $Q_{sr\tau}$ as the result of Hamilton product between $Q_s$ and $R_{r\tau}^\Delta$:

$$Q_{sr\tau} = Q_s \otimes R_{r\tau}^\Delta = a_{sr\tau} + e_{sr\tau}\mathbf{i} + f_{sr\tau}\mathbf{j} + g_{sr\tau}\mathbf{k}. \tag{18}$$

Finally, the scoring function of HTM is defined by the inner product:

$$\varphi(s, r, o, \tau)_{HTM} = Q_{sr\tau} \cdot Q_o. \tag{19}$$

## B.2. CTM for TKGC

This section follows the principles upon which we built CTM, we construct a model named CTM based on complex embeddings. For CTM, head entity $s$ and tail entity $o$ correspond to $C_s = \{\ a_s + e_s\mathbf{i} : a_s, e_s \in \mathbb{R}^k\ \}$ and $C_o = \{\ a_o + e_o\mathbf{i} : a_o, e_o \in \mathbb{R}^k\ \}$, respectively, while relation $r$ and timestamp $\tau$ correspond to $C_r = \{\ a_r + e_r\mathbf{i} : a_r, e_r \in \mathbb{R}^k\ \}$ and $C_\tau = \{\ a_\tau + e_\tau\mathbf{i} : a_\tau, e_\tau \in \mathbb{R}^k\ \}$, respectively. The specific steps are as follows:

We first normalize timestamp $C_\tau$ to $C_\tau^\Delta$ to eliminate the scaling effect by dividing $C_\tau$ by its norm:

$$C_\tau^\Delta = \frac{C_\tau}{|C_\tau|} = \frac{a_\tau + e_\tau\mathbf{i}}{\sqrt{a_\tau^2 + e_\tau^2}}. \tag{20}$$

Secondly, we get the temporal relation $C_{r\tau}$ by rotating $C_r$ through the product with $C_\tau^\Delta$:

$$C_{r\tau} = C_r \bullet C_\tau^\Delta = a_{r\tau} + e_{r\tau}\mathbf{i}, \tag{21}$$

Next, we define the $C_{sr\tau}$ as the result of product between $C_s$ and $C_{r\tau}^\Delta$:

$$C_{sr\tau} = C_s \bullet C_{r\tau}^\Delta = a_{sr\tau} + e_{sr\tau}\mathbf{i}, \tag{22}$$

where $\bullet$ is complex product. Finally, the scoring function of CTM is defined by inner product:

$$\varphi(s, r, o, \tau)_{CTM} = C_{sr\tau} \cdot C_o. \tag{23}$$

## B.3. CDS for TKGC

This section follows the overall concept of TeDS applied to complex embeddings, named CDS. For CDS, head entity $s$, relation $r$, and tail entity $o$ correspond to $C_s = \{ a_s + e_s\mathbf{i} : a_s, e_s \in \mathbb{R}^k \}$, $C_r = \{ a_r + e_r\mathbf{i} : a_r, e_r \in \mathbb{R}^k \}$, and $C_o = \{ a_o + e_o\mathbf{i} : a_o, e_o \in \mathbb{R}^k \}$, respectively. Timestamp $\tau$ is divided into diachronic timestamp $d\tau$ and synchronic timestamp $s\tau$, corresponding to $C_{d\tau}$ and $C_{s\tau}$, respectively. The specific steps are as follows:

**Synchronic perception (SP).** Firstly, to achieve a thorough integration of temporal and relational information, SP performs a composite reorganization of synchronic timestamp $C_{s\tau} = a_{s\tau} + e_{s\tau}\mathbf{i}$ and relation $C_r$. This results in the creation of two complex numbers: $C_{r\tau_{sp}} = a_r + e_{s\tau}\mathbf{i}$ and $C_{\tau_{sp}r} = a_{s\tau} + e_r\mathbf{i}$, where $a_{s\tau}, e_{s\tau} \in \mathbb{R}^k$. Secondly, we normalize $C_{\tau_{sp}r}$ to the unit complex $Q_{\tau_{sp}r}^{\Delta}$ by dividing $Q_{\tau_{sp}r}$ by its norm to eliminate scaling effects:

$$C_{\tau_{sp}r}^{\Delta} = \frac{C_{\tau_{sp}r}}{|C_{\tau_{sp}r}|} = \frac{a_\tau + e_\tau\mathbf{i}}{\sqrt{a_\tau^2 + e_\tau^2}}. \tag{24}$$

CDS utilizes complex product to perform rotation operations on $C_{r\tau_{sp}}$ via $C_{\tau_{sp}r}^{\Delta}$, obtaining representation of synchronic relations:

$$\mathcal{M}_c = C_{r\tau_{sp}} \bullet C_{\tau_{sp}r}^{\Delta}, \tag{25}$$

where $\bullet$ is complex product. Meanwhile, we normalize $\mathcal{M}_c$ to the unit complex $\mathcal{M}_c^{\Delta}$. We then rotate head entity $C_s$ by doing complex product between it and $\mathcal{M}_c^{\Delta}$:

$$C_{sp} = C_s \bullet \mathcal{M}^{\Delta}. \tag{26}$$

**Diachronic perception (DP).** Firstly, we iterate through all timestamps, sequentially mapping each time point to an angle and converting it to the scalar component of a complex number using trigonometric functions to construct diachronic timestamps:

$$C_{d\tau} = \cos(\frac{\pi \mathcal{T}_n}{2\mathsf{T}}) + \sin(\frac{\pi \mathcal{T}_n}{2\mathsf{T}})e'\mathbf{i}, \tag{27}$$

where $e' \in \mathbb{R}^k$; $\mathcal{T}_n \in \{1, ..., \mathsf{T}\}$ denotes time point. Secondly, we use complex product with the diachronic timestamp $C_{\tau_n}$ to rotate relation $C_r$ into diachronic relation $C_{r\tau_n}$:

$$C_{r\tau_n} = C_{\tau_n} \bullet C_r. \tag{28}$$

Thirdly, we normalize $C_{r\tau_n}$ to unit complex $C_{r\tau_n}^{\Delta}$. We then rotate the head entity $C_s$ by doing complex product between it and $C_{r\tau_n}^{\Delta}$:

$$C_{dp} = C_s \bullet C_{r\tau_n}^{\Delta}. \tag{29}$$

Finally, we summarize the scoring function of CDS as follows:

$$\varphi(s, r, o, \tau)_{CDS} = (C_{sp} + C_{dp}) \cdot C_o, \tag{30}$$

where $\cdot$ is inner product.

## B.4. Analysis of TeDS under unbalanced time distribution

Figure 11 extracts facts between *Barack Obama*, *Angela Merkel*, and *Russia* under different time spans from ICEWS14. Intuitively, DP and TeDS exhibit improved performance as the time span increases, while SP and HTM demonstrate a slight decline in performance with longer time spans. These results further support the notion that DP excels at capturing diachronic scenarios.

Figure 12 extracts different months from ICEWS14 to test temporal non-stationarity and synchronic scenarios. Intuitively, TeDS achieves significant and stable performance across all months, while DP consistently delivers the second-best results in different months. These results demonstrate that TeDS and DP excel at learning synchronic scenarios. Additionally, SP maintains good overall performance but experiences slight declines as diachronic scenarios are segmented. Overall, both DP and SP demonstrate stable and robust performance on strongly constrained datasets across different scenarios. This further supports our motivation: the two scenarios are not entirely independent but demonstrate mutual dependence and serve as foundations for each other.

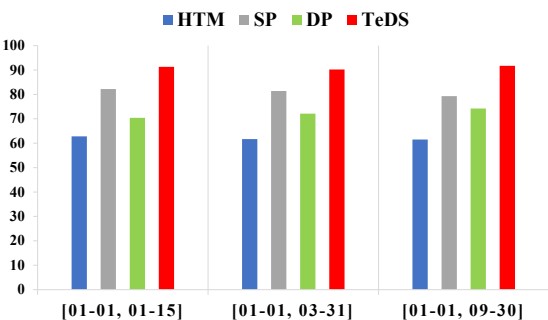

*Figure 11.* MRR performance comparison under different time spans on ICEWS14. Where [*01-01*, *01-15*] represents the years *2014-1-1* to *2014-1-15*; [*01-01*, *03-31*] represents the years *2014-1-1* to *2014-3-31*; [*01-01*, *09-30*] represents the years *2014-1-1* to *2014-9-30*.

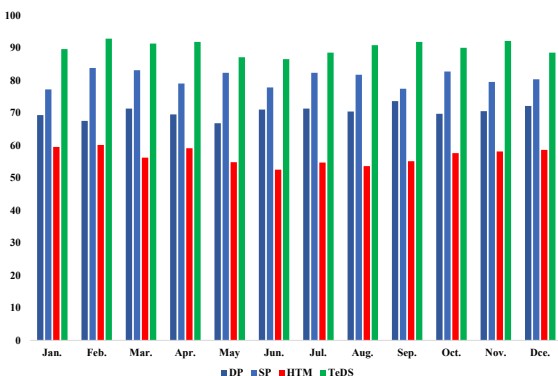

*Figure 12.* Comparison of MRR Performance Indicators for HTM, SP, DP, and TeDS Based on Different Monthly Data from ICEWS14.

Furthermore, we find that TeDS outperforms TPComplEx on datasets with longer time spans, which we attribute to its focus on synchronicity and diachronicity. In contrast, TPComplEx is limited to a synchronic perspective, overlooking the impact of facts over time. To validate our hypothesis, we test TeDS and TPComplEx on a long-time-span constrained dataset concerning *Obama* and *Japan*, as shown in Figure 13. The results consistently show that TeDS outperforms TPComplEx, further supporting our hypothesis.

### B.5. Performance comparison between TPComplEx and TeDS

Compared to TPComplEx, TeDS demonstrates significantly better performance in MRR and Hit@1 but slightly lower performance in Hit@3 and Hit@10 on ICEWS14. On ICEWS05-15, TeDS shows significantly better performance in MRR, Hit@1, and Hit@3, while slightly underperforming in Hit@10. Similarly, on ICEWS18, TeDS achieves significantly better performance in MRR, Hit@1, and Hit@3, but its performance in Hit@10 is slightly lower. Overall, TeDS performs comparably to TPComplEx on ICEWS series datasets. The key difference lies in the fact that TeDS's performance primarily benefits from its exploration of synchronicity and diachronicity in temporal properties, with a focus on selecting the unique optimal answer. In contrast, TPComplEx's performance is mainly derived from its study of simultaneity, aggregation, and associativity in temporal properties, emphasizing the aggregation and simultaneous prediction of multiple facts, thereby focusing more on correctly predicting all possible correct answers within the ranking range. In fact, balancing the selection of the optimal answer while accounting for multiple correct answers within the ranking range is an intriguing research topic.

For a comprehensive comparison, we reproduce TPComplEx (selecting best result from rank ∈ {1000, 1500, 2000}) on Wikidata12k, YAGO11k, and ICEWS18. Table 5 and 6 show that TeDS outperforms TPComplEx across all datasets, with significant improvements on Wikidata12k and YAGO11k, which we attribute to TeDS's modeling of temporal scenarios.

Finally, we perform a computational complexity comparison between TPComplEx and TeDS. Table 7 shows TeDS's superior training speed: 43% faster per epoch on Wikidata12k (5.63s vs. 9.87s) and 35% faster on YAGO11k (2.82s vs. 4.33s)

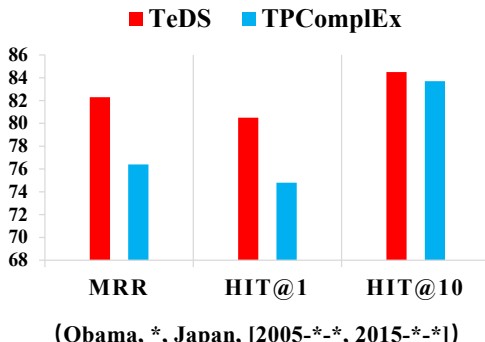

**(Obama, *, Japan, [2005-\*-\*, 2015-\*-\*])**

*Figure 13.* Performance comparison of the strongly constrained test between Obama and Japan on ICEWS05-15.

*Table 5.* Experimental results on YAGO11k, Wikidata12k, and GDELT. All results are taken from the original papers. Dashes: results are not reported in the responding literature. The best results among all models are written bold.

|  | YAGO11k | | | | Wikidata12k | | | | GDELT | | | |
|---|---|---|---|---|---|---|---|---|---|---|---|---|
|  | MRR | H@1 | H@3 | H@10 | MRR | H@1 | H@3 | H@10 | MRR | H@1 | H@3 | H@10 |
| TPComplEx | 38.5 | 29.4 | 42.6 | 58.6 | 42.1 | 31.5 | 45.7 | 66.1 | 40.7 | 32.9 | 43.1 | 55.9 |
| CDS (ours) | 58.1 | 57.4 | 58.7 | 60.8 | 70.7 | 70.3 | 71.9 | 74.1 | 58.6 | 56.1 | 59.0 | 61.5 |
| TeDS (ours) | 77.3 | 76.9 | 78.1 | 78.4 | 95.4 | **95.5** | **95.7** | 95.8 | 71.5 | 69.9 | 72.2 | 74.4 |
| DTeDS (ours) | **81.0** | **80.3** | **81.1** | **81.8** | **95.7** | 94.6 | 95.4 | **96.0** | **85.6** | **82.8** | **86.4** | 90.0 |

compared to TPComplEx. Crucially, these speedups are achieved alongside dramatic parameter reductions (e.g., 90% fewer parameters on YAGO11k). The marginally longer runtime on ICEWS05-15 (30.75s vs. 22.06s) is justified by TeDS using only 27% of baseline's parameters—a favorable tradeoff for memory-constrained applications. Table 8 reveals TeDS's most striking advantage: achieving superior performance with just 20.64M parameters versus TPComplEx's 201.2M on Wikidata12k—a 10× improvement in parameter efficiency.

The above analysis demonstrates that TeDS is particularly well-suited for large-scale TKG applications, maintaining competitive performance while significantly reducing memory overhead and computational resource requirements

### B.6. Hyperparameter analysis

Figure 14 illustrates the complete hyperparameter configuration of TeDS, where $\lambda_a$ is the regularization weight and $\lambda_b$ is the temporal regularization weight. Our observations indicate that $\lambda_a$ impacts results more than $\lambda_b$.

Moreover, we additionally plot the loss convergence curves of TeDS on ICEWS14 and ICEWS05-15, which demonstrate its stable and efficient optimization process.

### B.7. TeDS can model the symmetry/antisymmetry pattern.

**Theorem B.1.** *Relation $r$ is (anti)symmetric if $\forall x, y, \tau$*

$$(x, r, y, \tau) \Rightarrow (y, r, x, \tau), (x, r, y, \tau) \Rightarrow \neg(y, r, x, \tau). \tag{31}$$

*A fact with such form is a **(anti)symmetric pattern**.*

The flexibility and representational power of quaternion enable us to model various relation patterns at ease.

*Proof.* For symmetric pattern, if $r(s, o, \tau)$ and $r(o, s, \tau)$ hold, we have:

$$Q_s \otimes (R_{r\tau_n}^{\Delta} + \mathscr{M}^{\Delta}) \cdot Q_o = Q_o \otimes (R_{r\tau_n}^{\Delta} + \mathscr{M}^{\Delta}) \cdot Q_s. \tag{32}$$

*Table 6.* Experimental results on ICEWS14, ICEWS05-15, and ICEWS18. All results are taken from the original papers. Dashes: results are not reported in the responding literature. The best results among all models are written bold.

| | ICEWS14 | | | | ICEWS05-15 | | | | ICEWS18 | | | |
| --- | --- | --- | --- | --- | --- | --- | --- | --- | --- | --- | --- | --- |
| | MRR | H@1 | H@3 | H@10 | MRR | H@1 | H@3 | H@10 | MRR | H@1 | H@3 | H@10 |
| TPComplEx (2024) | 89.8 | 86.5 | 92.5 | 95.4 | 84.5 | 79.4 | 88.2 | 93.4 | 34.2 | 23.7 | 31.6 | 45.3 |
| CDS (ours) | 83.1 | 82.4 | 83.7 | 85.8 | 75.4 | 69.5 | 74.8 | 78.1 | 32.5 | 27.1 | 35.5 | 47.1 |
| TeDS (ours) | 90.7 | 90.6 | 90.8 | 91.0 | 87.3 | 85.8 | 88.3 | 90.4 | 45.8 | 45.6 | 46.9 | 48.2 |
| DTeDS (ours) | **92.1** | **91.4** | **92.1** | **92.6** | **95.4** | **90.1** | **91.0** | **92.9** | **56.6** | **55.4** | **58.8** | **60.6** |

*Table 7.* Training Time Comparison: TeDS vs. TPComplEx (seconds per epoch).

| Model | ICEWS14 | ICEWS05-15 | Wikidata12k | YAGO11k |
| --- | --- | --- | --- | --- |
| TPComplEx | 5.56 | 22.06 | 9.87 | 4.33 |
| TeDS(ours) | 6.30 | 30.75 | 5.63 | 2.82 |

*Table 8.* Parameter Count Comparison: TeDS vs. TPComplEx.

| Model | ICEWS14 | ICEWS05-15 | Wikidata12k | YAGO11k |
| --- | --- | --- | --- | --- |
| TPComplEx | 92,362,736 | 75,228,488 | 201,260,000 | 170,140,000 |
| TeDS(ours) | 12,065,600 | 20,396,800 | 20,643,200 | 17,340,000 |

The symmetry of TeDS can be demonstrated by setting the imaginary part of $R_{r\tau_n}^\Delta + \mathcal{M}^\Delta$ to 0. Since $R_{r\tau_n} = W_{d\tau_n} \otimes R_r$ and $\mathcal{M} = Q_{r\tau_{sp}} \otimes Q_{\tau_{sp}r}^\Delta$, according to Hamilton product rule, both $R_r = 0$ and $R_{r\tau}^\Delta = 0$ can be achieved by setting the imaginary parts of $R_r$ and $Q_{\tau_{sp}r}^\Delta$ to 0.

For antisymmetric pattern, if $r(s, o, \tau)$ and $\neg r(o, s, \tau)$ hold, we have:

$$Q_s \otimes (R_{r\tau_n}^\Delta + \mathcal{M}^\Delta) \cdot Q_o \neq Q_o \otimes (R_{r\tau_n}^\Delta + \mathcal{M}^\Delta) \cdot Q_s. \tag{33}$$

The antisymmetry of TeDS can be demonstrated by ensuring that the imaginary parts of $R_{r\tau_n}^\Delta + \mathcal{M}^\Delta$ are nonzero.

$\square$

## B.8. Proof of Lemma 2

**Theorem B.2.** *Relation $r$ is **inverse** to relation $r^{-1}$ if $\forall x, y, \tau$*

$$(x, r, y, \tau) \Rightarrow (y, r^{-1}, x, \tau). \tag{34}$$

*A fact with such form is a **inverse pattern**.*

*Proof.* For inverse pattern, if $r_1(s, o, \tau)$ and $r_2(o, s, \tau)$ hold, we have:

$$Q_s \otimes (R_{r_1\tau_n}^\Delta + \mathcal{M}^\Delta) \cdot Q_o = Q_o \otimes (R_{r_2\tau_n}^\Delta + \mathcal{M}^\Delta) \cdot Q_s. \tag{35}$$

For inverse property of TeDS, we need to utilize the conjugation of quaternions to prove the case since conjugation of the quaternion is its own inverse. Thus, formula 35 can be rewritten in the following form:

$$\begin{aligned}
& Q_s \otimes (R_{r\tau_n}^\Delta + \mathcal{M}^\Delta) \cdot Q_o = Q_o \otimes \overline{(R_{r\tau_n}^\Delta + \mathcal{M}^\Delta)} \cdot Q_s \\
\Longleftrightarrow & Q_s \otimes R_{r\tau_n}^\Delta \cdot Q_o + Q_s \otimes \mathcal{M}^\Delta \cdot Q_o = Q_o \otimes \overline{R_{r\tau_n}^\Delta} \cdot Q_s + Q_o \otimes \overline{\mathcal{M}^\Delta} \cdot Q_s \\
\Longrightarrow & Q_s \otimes R_{r\tau_n}^\Delta \cdot Q_o = Q_o \otimes \overline{R_{r\tau_n}^\Delta} \cdot Q_s; Q_s \otimes \mathcal{M}^\Delta \cdot Q_o = Q_o \otimes \overline{\mathcal{M}^\Delta} \cdot Q_s
\end{aligned} \tag{36}$$

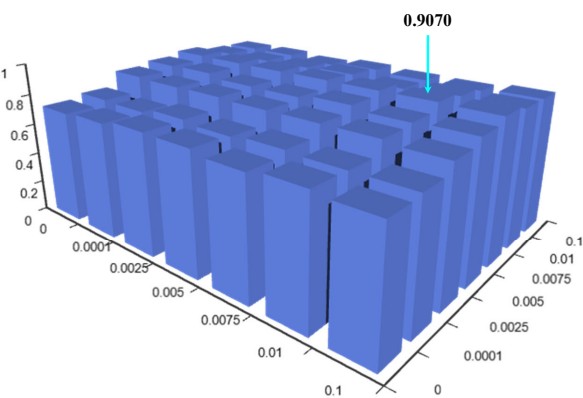

*Figure 14.* Performance of MRR with different $\lambda_a$ and $\lambda_b$ on ICEWS14.

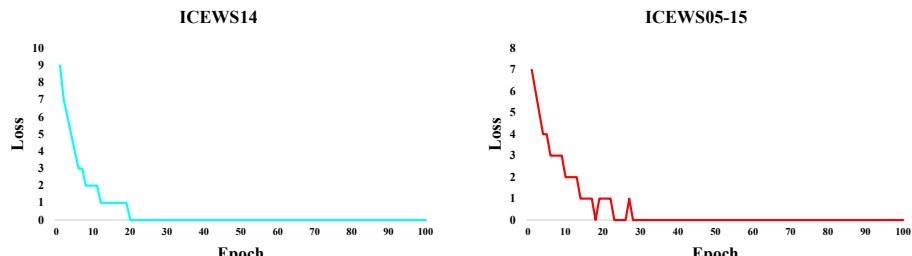

*Figure 15.* Convergence curves of TeDS on ICEWS14 and ICEWS05-15.

To prove $Q_s \otimes \mathscr{M}^\Delta \cdot Q_o = Q_o \otimes \overline{\mathscr{M}^\Delta} \cdot Q_s$, we first have:

$$
\begin{aligned}
\mathscr{M} = Q_{r\tau_{sp}} \otimes Q_{\tau_{sp}r} = & (a_r a_{s\tau} - e_r e_{s\tau} - f_r f_{s\tau} - g_{s\tau} g_r) + \\
& (a_r e_{s\tau} + e_r a_{s\tau} + f_r g_{s\tau} - g_{s\tau} f_r)\mathbf{i} + \\
& (a_r f_{s\tau} - e_r g_{s\tau} + f_r a_{s\tau} + g_{s\tau} e_r)\mathbf{j} + \\
& (a_r g_{s\tau} + e_r f_{s\tau} - f_r e_{s\tau} + g_{s\tau} a_r)\mathbf{k}.
\end{aligned}
\tag{37}
$$

We let $a_{rs\tau} = a_r a_{s\tau} - e_r e_{s\tau} - f_s f_{s\tau} - g_{s\tau} g_r$, $e_{rs\tau} = a_r e_{s\tau} + e_r a_{s\tau} + f_r g_{s\tau} - g_{s\tau} f_r$, $f_{rs\tau} = a_r f_{s\tau} - e_r g_{s\tau} + f_r a_{s\tau} + g_{s\tau} e_r$, $g_{rs\tau} = a_r g_{s\tau} + e_r f_{s\tau} - f_r e_{s\tau} + g_{s\tau} a_r$. We normalize $\mathscr{M}$ to unit quaternion $\mathscr{M}^\Delta$:

$$
\mathscr{M}^\Delta(a'', e'', f'', g'') = \frac{\mathscr{M}}{|\mathscr{M}|} = \frac{a_{rs\tau} + e_{rs\tau}\mathbf{i} + f_{rs\tau}\mathbf{j} + g_{rs\tau}\mathbf{k}}{\sqrt{a_{rs\tau}^2 + e_{rs\tau}^2 + f_{rs\tau}^2 + g_{rs\tau}^2}}.
\tag{38}
$$

Next, we obtain the conjugate of $\mathscr{M}^\Delta$:

$$
\overline{\mathscr{M}^\Delta} = a'' - e''\mathbf{i} - f''\mathbf{j} - g''\mathbf{k}.
\tag{39}
$$

And then, we expand the left term of formula (35):

$$
\begin{aligned}
&Q_s \otimes \mathscr{M}^\Delta \cdot Q_o \\
=&[(a_s a'' - e_s e'' - f_s f'' - g_s g'')+ \\
&(a_s e'' + e_s a'' + f_s g'' - g_s f'')\mathbf{i}+ \\
&(a_s f'' - e_s g'' + f_s a'' + g_s e'')\mathbf{j}+ \\
&(a_s g'' + e_s f'' - f_s e'' + g_s a'')\mathbf{k}] \\
&\cdot (a_o + e_o\mathbf{i} + f_o\mathbf{j} + g_o\mathbf{k}). \\
=&(a_s a'' - e_s e'' - f_s f'' - g_s g'') \cdot a_o+ \\
&(a_s e'' + e_s a'' + f_s g'' - g_s f'') \cdot e_o+ \\
&(a_s f'' - e_s g'' + f_s a'' + g_s e'') \cdot f_o+ \\
&(a_s g'' + e_s f'' - f_s e'' + g_s a'') \cdot g_o \\
=&\langle a_s, a'', a_o \rangle - \langle e_s, e'', a_o \rangle - \langle f_s, f'', a_o \rangle - \langle g_s, g'', a_o \rangle + \\
&\langle a_s, e'', e_o \rangle + \langle e_s, a'', e_o \rangle + \langle f_s, g'', e_o \rangle - \langle g_s, f'', e_o \rangle + \\
&\langle a_s, f'', f_o \rangle - \langle e_s, g'', f_o \rangle + \langle f_s, g'', f_o \rangle + \langle g_s, e'', f_o \rangle + \\
&\langle a_s, g'', g_o \rangle + \langle e_s, f'', g_o \rangle - \langle f_s, e'', g_o \rangle + \langle g_s, a'', g_o \rangle .
\end{aligned}
\tag{40}
$$

Finally, we expand the right term of formula (35):

$$
\begin{aligned}
&Q_o \otimes \overline{\mathscr{M}^\Delta} \cdot Q_s \\
=&[(a_o a'' + e_o e'' + f_o f'' + g_o g'')+ \\
&(-a_o e'' + e_o a'' - f_o g'' + g_o f'')\mathbf{i}+ \\
&(-a_o f'' + e_o g'' + f_o a'' - g_o e'')\mathbf{j}+ \\
&(-a_o g'' - e_o f'' + f_o e'' + g_o a'')\mathbf{k}] \\
&\cdot (a_s + e_s\mathbf{i} + f_s\mathbf{j} + g_s\mathbf{k}). \\
=&(a_o a'' + e_o e'' + f_o f'' + g_o g'') \cdot a_s+ \\
&(-a_o e'' + e_o a'' - f_o g'' + g_o f'') \cdot e_s+ \\
&(-a_o f'' + e_o g'' + f_o a'' - g_o e'') \cdot f_s+ \\
&(-a_o g'' - e_o f'' + f_o e'' + g_o a'') \cdot g_s \\
=&\langle a_o, a'', a_s \rangle + \langle e_o, e'', a_s \rangle + \langle f_o, f'', a_s \rangle + \langle g_o, g'', a_s \rangle - \\
&\langle a_o, e'', e_s \rangle + \langle e_o, a'', e_s \rangle - \langle f_o, g'', e_s \rangle + \langle g_o, f'', e_s \rangle - \\
&\langle a_o, f'', f_s \rangle + \langle e_o, g'', f_s \rangle + \langle f_o, g'', f_s \rangle - \langle g_o, e'', f_s \rangle - \\
&\langle a_o, g'', g_s \rangle - \langle e_o, f'', g_s \rangle + \langle f_o, e'', g_s \rangle + \langle g_o, a'', g_s \rangle .
\end{aligned}
\tag{41}
$$

By comparison, we can easily check the equality of left and right term. Similarly, we can derive that $Q_s \otimes R^\Delta_{r\tau_n} \cdot Q_o = Q_o \otimes \overline{R^\Delta_{r\tau_n}} \cdot Q_s$. Therefore, TeDS has sufficient capability to model inverse patterns.

$\square$

