# OpenReview forum: "TeDS: Joint Learning of Diachronic and Synchronic Perspectives in Quaternion Space for Temporal Knowledge Graph Completion"
_ICML.cc/2025/Conference — ICML 2025 poster_

### Official Review · Reviewer_rpDA · 2025-03-09

**Overall Recommendation:** 3

**Summary:**

This paper proposes TeDS, a temporal knowledge graph completion method cosidering both diachronicic and synchronic flows within and between temporalized facts. Extensive experiments demonstrate that TeDs is capable of achieving state-of-the-art performance across multiple benchmarks.

**Claims And Evidence:**

The performance improment compared with baselines is more marginal on the first three datasets (Table 2), could the authors explain this phenomenon?

**Essential References Not Discussed:**

It may exists some related works this paper hasn't included.

**Experimental Designs Or Analyses:**

TPComplEx seems to be a very strong baseline, so could the authors use its source code to produce results on the remaining three datasets? It would be much helpful to make a fairer comparison.

**Methods And Evaluation Criteria:**

Yes, the paper is easy to follow, with clear writing and well-presented technical details.

**Other Comments Or Suggestions:**

The font-size of most figures is overly small.

**Other Strengths And Weaknesses:**

In Section 3.2, the author may briefly introduce Quaternion and Hamilton rule with examples for readibility.

**Questions For Authors:**

Please see above comments.

**Relation To Broader Scientific Literature:**

Modeling TKGC problem from both diachronicic and synchronic perspectives is novel and highly motivated. The usage of quaternion theroies is also interesting.

**Theoretical Claims:**

I didn't  check all the proofs.

---

> ### Author Rebuttal · Authors · 2025-03-31
>
> Thanks for all your valuable comments. Note: The pictures and tables used in response are available at https://anonymous.4open.science/r/TEDS-033A/To_Re_rpDA.pdf See: To_Re_rpDA.pdf
>
> Q1: The performance improment compared with baselines is more marginal on first three datasets (Table2), could the authors explain this phenomenon.
>
> A1: TeDS gets consistent performance improvements across three benchmark datasets—ICEWS14, ICEWS05-15, and ICEWS18—with particularly notable enhancements on ICEWS18. This disparity primarily stems from ICEWS18 containing approximately ten times more facts than ICEWS05-15 at the same temporal density, significantly increasing the complexity of temporal context modeling. By effectively capturing intricate dependencies among high-density facts through its Synchronous Perception (SP) mechanism, TeDS attains optimal performance on this dataset.
>
> Furthermore, TeDS demonstrates even more pronounced advantages on larger-scale TKGs with greater challenges, including YAGO11k, Wikidata12k, and GDELT. For YAGO11k and Wikidata12k, which feature long temporal spans and distinct long-tail distributions, TeDS's Diachronic Perception (DP) employs dynamic smooth embedding techniques to simultaneously model short-term fluctuations and long-term evolutionary trends, effectively addressing the challenges of sparse temporal data modeling. Meanwhile, on the extremely fact-dense GDELT dataset, TeDS significantly enhances capture of complex temporal patterns by collaboratively optimizing diachronic and synchronous feature representations. These comparative results fully demonstrate TeDS's unique strengths in handling high-density facts and complex temporal dependencies, with its synchronous and diachronic perception capabilities being more fully realized.
>
>
> Q2: TPComplEx seems to be a very strong baseline, so could the authors use its source code to produce results on remaining three datasets? It would be much helpful to make a fairer comparison.
>
> A2: To compare, we reproduce TPComplEx (selecting best result from rank $\in$ {1000, 1500, 2000}) on Wikidata12k, YAGO11k, and ICEWS18. Table 1 and 2 show that TeDS outperforms TPComplEx across all datasets, with significant improvements on Wikidata12k and YAGO11k, which we attribute to TeDS's modeling of temporal scenarios.
>
> Besides, we compare two under strongly constrained datasets in the Appendix of our manuscript (B.6. Performance comparison between TPComplEx and TeDS).
>
> Finally, we perform a computational complexity comparison between TPComplEx and TeDS. Table 3 shows TeDS's superior training speed: 43% faster per epoch on Wikidata12k (5.63s vs. 9.87s) and 35% faster on YAGO11k (2.82s vs. 4.33s) compared to TPComplEx. Crucially, these speedups are achieved alongside dramatic parameter reductions (e.g., 90% fewer parameters on YAGO11k). The marginally longer runtime on ICEWS05-15 (30.75s vs. 22.06s) is justified by TeDS using only 27% of baseline's parameters—a favorable tradeoff for memory-constrained applications. Table 4 reveals TeDS's most striking advantage: achieving superior performance with just 20.64M parameters versus TPComplEx's 201.2M on Wikidata12k—a 10× improvement in parameter efficiency.
>
> The above analysis demonstrates that TeDS is particularly well-suited for large-scale TKG applications, maintaining competitive performance while significantly reducing memory overhead and computational resource requirements
>
>
> Q3: It seems the authors did not upload their source code and did not commit to releasing the code after acceptance.
>
> A3: All source code used for conducting and analyzing the experiments will be publicly available upon the publication of the paper under a license that permits free use for research purposes.
>
>
> Q4: It may exists some related works this paper hasn't included.
>
> A4: We add more important references to our paper (See our response to Reviewer C2mh's A2 for details). We continue collecting latest (2025) references (e.g., MTE, Neo-TKGC, and GLARGCN) to further validate TeDS's superiority (Table 5 and 6).
>
>
> Q5: In Section 3.2, the author may briefly introduce Quaternion and Hamilton rule with examples for readibility.
>
> A5: Based on your detailed comments, we have added the following content: Quaternion is a prominent example of hypercomplex number system, extending the traditional complex number system into four-dimensional space. A quaternion $Q$ consists of one real component and three imaginary components, defined as $Q=a + e \mathbf{i} + f \mathbf{j} + g \mathbf{k}$, where $a, e, f, g$ are real numbers, $\mathbf{i}, \mathbf{j}, \mathbf{k}$ are imaginary units, satisfying Hamilton rule:
>
> 1. $\mathbf{i}^{2} = \mathbf{j}^{2} = \mathbf{k}^{2}=-1$
>
> 2. $ij=k, ji=-k$
>
> 3. $jk=i, kj=-i$
>
> 4. $ki=j, ik=-j$
>
> Q6: The font-size of most figures is overly small.
>
> A6: We have reviewed our manuscript and accepted your suggestions. In revised version, we will adjust the font size of the figures to enhance the readability of our manuscript.

---

### Official Review · Reviewer_WGeZ · 2025-03-11

**Overall Recommendation:** 3

**Summary:**

This paper introduces TeDS, which is a unified framework to simultaneously consider both diachronic timestamp and synchronic timestamp for TKGC. TeDS achieves significant improvements over the existing SOTA on six datasets.

**Claims And Evidence:**

The effectiveness of modeling TKGs from the perspectives of synchronic perception and diachronic perception has been validated in the experimental results.

**Essential References Not Discussed:**

Please see Weakness 1:
Some approaches of mapping diachronicity and synchronicity to Quaternion Space for modeling are missing:
[1] Combination of translation and rotation in dual quaternion space for temporal knowledge graph completion, IJCNN 2023
[2] TELS: Learning time-evolving information and latent semantics using dual quaternion for temporal knowledge graph completion, KBS 2024

**Experimental Designs Or Analyses:**

I have checked the soundness and validity of the experimental designs and analyses.

**Methods And Evaluation Criteria:**

The proposed methods and evaluation criteria are appropriate for the TKGC problem.

**Other Comments Or Suggestions:**

No

**Other Strengths And Weaknesses:**

Strength:
The experiments are relatively comprehensive, and TeDS achieves the state-of-the-art performance.

Weakness:
1. The concepts of diachronic timestamp and synchronic timestamp appear to correspond to the temporal and structural dependencies in TKGs, respectively, which are fundamental considerations in most TKGs studies. The approach of mapping these dependencies to Quaternion Space for modeling is reasonable. However, there are notable similarities between this paper and ComTR [1] in terms of structure and equations, with the exception of the Diachronic Perception component. Despite this, ComTR is not cited or discussed, which seems inappropriate. The authors should explicitly clarify the differences and similarities between their work and ComTR. In addition, TELS [2] also adopts Quaternion Space for modeling TKGs, and a comparison with it would further strengthen the paper.
[1] Combination of translation and rotation in dual quaternion space for temporal knowledge graph completion, IJCNN 2023
[2] TELS: Learning time-evolving information and latent semantics using dual quaternion for temporal knowledge graph completion, KBS 2024

2. The Related Work section would be more informative if it included a clearer discussion on the relevance of previous studies to TeDS.

3. The visualizations of the temporal relation embeddings should include visualizations results from some baseline methods, rather than only comparisons with variations of TeDS.

4. Adding a discussion on the limitations of the study and potential future directions would enhance the completeness of the paper.

5. The text in most figures is not easily readable. Enhancing the clarity of the images would improve overall readability and presentation quality.

**Questions For Authors:**

No

**Relation To Broader Scientific Literature:**

Considering both diachronicity and synchronicity is beneficial for TKGC.

**Theoretical Claims:**

I have checked the correctness of the proofs for the theoretical claims.

---

> ### Author Rebuttal · Authors · 2025-03-31
>
> Thanks for all your valuable comments. The pictures and tables used in response are available at https://anonymous.4open.science/r/TEDS-033A/To_Re_WGeZ.pdf See: To_Re_WGeZ.pdf
>
> Q1: The concepts of diachronic timestamp and synchronic timestamp appear to correspond to the temporal and structural dependencies in TKGs, respectively, which are fundamental considerations in most TKGs studies. The approach of mapping these dependencies to Quaternion Space for modeling is reasonable. However, there are notable similarities between this paper and ComTR in terms of structure and equations, with the exception of Diachronic Perception component. Despite this, ComTR is not cited or discussed, which seems inappropriate. The authors should explicitly clarify the differences and similarities between their work and ComTR. Besides, TELS also adopts Quaternion Space for modeling TKGs, and a comparison with it would further strengthen paper
>
> The related work would be more informative if it included a clearer discussion on relevance of previous studies to TeDS
>
> A1: We add above references and provide following response
>
> 1)Differences in underlying technical details: TeDS uses quaternions, while ComTR uses dual quaternions. In SP, TeDS achieves a thorough integration of temporal and relational information by reorganizing the synchronous timestamp $ W_{s\tau} = a_{s\tau} + e_{s\tau} \mathbf{i} + f_{s\tau} \mathbf{j} + g_{s\tau} \mathbf{k} $ and relation $ R_r $, forming two quaternions $ Q_{r\tau_{sp}} $ and $ Q_{\tau_{sp}r} $. In contrast, ComTR directly applies dual quaternions without deep information interaction. Details in Section 4 (TeDS for TKGC).
>
> 2)Different motivations: TeDS observes regularities of facts within a temporal context and summarizes two important temporal perspectives: synchronicity and diachronicity. In contrast, ComTR focuses on capturing multi-relational patterns in a temporal context.
>
> 3)TeDS integrates dual perception channels into a unified framework to handle multi-perspective temporal facts. It is not limited to quaternions and adapts to tensors, complex numbers, dual quaternions, and homogeneous transformations. In contrast, ComTR relies on rule modeling based on dual quaternion properties.
>
> Like ComTR, TELS uses dual quaternions to: 1)model multiple relations pattern; 2)model evolutionary hierarchical pattern; 3) capture unique latent semantics based on an entity's position in a relation. Unlike ComTR, TELS maximizes dual quaternion advantages rather than just applying them. Besides, TELS components become more portable and less reliant on dual quaternion technology (e.g., latent semantic and evolutionary hierarchical awareness).
>
> To show TeDS's portability and effectiveness, we implement it with dual quaternions as DTeDS (see Table1 and 2). Compared to baselines, DTeDS and TeDS consistently get best and second-best results. We also observe that DTeDS and TELS outperform TeDS on GDELT. Due to GDELT's high data density at same temporal granularity, many multi-relation patterns emerge. DTeDS and TELS leverage the strength of dual quaternions in capturing these patterns, aligning with our hypothesis. Besides, we implement TeDS with complex numbers as CDS, which also shows competitive results. This further confirms the portability and effectiveness of our framework.
>
>
>
> Q2: The visualizations of temporal relation embeddings should include visualizations results from some baseline methods, rather than only comparisons with variations of TeDS.
>
> A2: We enhance comparison by visualizing TeLM’s temporal relation embeddings. First, we examine distribution of same relation over time, extracting relation Consult between Obama and Netanyahu in 2014 (Fig 1). TeLM outperforms HTM in classifying Consult across months, but is less effective than TeDS in aggregating data from multiple months, with blurred boundaries between adjacent months. Next, we observe various relations between Obama and Netanyahu from Jan-Jun 2014 (Fig 2). TeDS and SP perform better than HTM and TeLM in distinguishing relations. TeDS surpasses SP in differentiating identical relations, rather than clustering them together. Even within the same month, relations may show different trends based on context. Thus, clustering different relations while preserving uniqueness of identical relations in specific contexts is key.
>
>
>
> Q3: Adding a discussion on limitations of study and potential future directions would enhance completeness of paper.
>
> A3: Our TKG research focuses on factual records, which lack clear cycles like seasons or biological rhythms. Modeling cycles is key for prediction. We may use Fourier transforms or seasonal decomposition to enhance TeDS.
>
>
>
> Q4: The text in most figures is not easily readable. Enhancing the clarity of images would improve overall readability and presentation quality.
>
> A4: We will revise paper structure and adjust the font size in figures in revised version to enhance readability of manuscript.

---

### Official Review · Reviewer_fTHA · 2025-03-13

**Overall Recommendation:** 2

**Summary:**

The paper introduces a quaternion-based model for temporal knowledge graph completion that integrates diachronic and synchronic perspectives. The model demonstrates significant improvements over state-of-the-art methods on six benchmark datasets, showcasing its effectiveness in handling both short and long temporal spans.

**Claims And Evidence:**

The authors provide comprehensive experiments on multiple datasets, showing that TeDS outperforms existing methods in terms of metrics such as Mean Reciprocal Rank (MRR) and Hits@n. The ablation studies further validate the effectiveness of the proposed dual perception channels. However, a more detailed discussion on the computational complexity and scalability of the model would strengthen the claims.

**Essential References Not Discussed:**

No critical omissions noted

**Experimental Designs Or Analyses:**

The authors conducted experiments on multiple benchmark datasets, including ICEWS, YAGO11k, and Wikidata12k, demonstrating the robustness of TeDS across different temporal scenarios. The ablation studies provide insights into the contributions of the synchronic and diachronic perception channels.

**Methods And Evaluation Criteria:**

The proposed methods and evaluation criteria are appropriate for the problem. The use of quaternion embeddings and the dual temporal perception channels are innovative approaches for temporal knowledge graph completion. The evaluation metrics (MRR, Hits@n) are standard in the field and suitable for assessing the model's performance.

**Other Comments Or Suggestions:**

1. Figure 3’s time distribution analysis could include more datasets.

2. Clarify whether TeDS can handle time intervals (e.g., [start, end]) beyond points.

**Other Strengths And Weaknesses:**

Strengths:

1. The dual temporal perception channels are an effective approach for capturing temporal dynamics.

2. Using quaternion embeddings provides a unique way to integrate temporal and relational information.

Weaknesses:

1. The paper lacks a detailed discussion on the computational complexity and scalability of the model.

2. Limited discussion of computational overhead compared to simpler models (e.g., TransE variants).

3. SP and DP modules underperform TeDS, but the combination’s superiority is attributed to "deep integration" without mechanistic explanation.

4. Quaternion operations introduce complexity without clear advantages over other methods.

**Questions For Authors:**

1. How does TeDS handle extremely sparse temporal data, and are there any limitations in such scenarios?

2. How does TeDS generalize to timestamps not seen during training (e.g., future events)?

**Relation To Broader Scientific Literature:**

The paper builds on prior work in quaternion embeddings (e.g., QuatE) and extends it to temporal knowledge graphs. The dual temporal perception channels address limitations in existing methods that often treat temporal information as supplementary.

**Theoretical Claims:**

The paper does not present extensive theoretical claims beyond the quaternion-based representation and its application to temporal knowledge graphs. The correctness of the quaternion operations and their application to temporal reasoning appears sound, but a deeper theoretical analysis (e.g., convergence properties, bounds on performance) would be beneficial.

---

> ### Author Rebuttal · Authors · 2025-03-31
>
> Thanks for all your valuable comments. Note: The pictures and tables used in response are available at https://anonymous.4open.science/r/TEDS-033A/To_Re_fTHA.pdf See: To_Re_fTHA.pdf
>
> Q1: A more detailed discussion on model's computational complexity and scalability would strengthen the claims. The paper lacks in-depth analysis, particularly regarding computational overhead compared to simpler models (e.g., TransE variants)
>
> A1:
>
> (1)Complexity Comparison. In Table1, TeDS has same theoretical complexity as mainstream models (space: $\mathcal{O}(n_ed + n_rd + n_td)$, time: $\mathcal{O}(d)$). Compared to quaternion models (RotateQVS and EHPR, rank=2000), TeDS gets best results with rank=100, reducing dimensionality by 80-95%, improving storage and computation efficiency while maintaining theoretical completeness.
>
> (2)Training Speed Comparison. Table2 shows TeDS's superior training speed.
>
> (3)Parameter Comparison. Table3 compares actual parameter counts.
>
> For (2) and (3), see our response to Reviewer cu6z's A6 for details TeDS ensures optimal performance while significantly reducing computational overhead, making it ideal for large-scale TKGs.
>
>
> Q2: The correctness of the quaternion operations and their application to temporal reasoning appears sound, but a deeper theoretical analysis (e.g., convergence properties, bounds on performance) would be beneficial
>
> A2: We add experiments: 1) loss function convergence curves on ICEWS14 and ICEWS05-15 (see Fig1); 2) hyperparameter sensitivity analysis (see Fig2); 3) embedding dimension analysis (See Fig3). (See our response to Reviewer cu6z's A3 for details)
>
>
> Q3: SP and DP modules underperform TeDS, but combination’s superiority is attributed to "deep integration" without mechanistic explanation
>
> A3: For SP and DP, we further analyze characteristics between modules through temporal relation embedding visualization (Section 6.1) and strong constraint experiments (Section 6.2). Besides, we visualize and compare temporal relation embedding of baseline TeLM (see our response to Reviewer WGeZ's A2 for details).
>
> Q4: Quaternion operations introduce complexity without clear advantages over other methods
>
> A4: We add latest baselines (e.g., MTE(2025), Neo-TKGC(2025), GLARGCN(2025)) to continuously evaluate TeDS's performance (Table4 and 5). Compared to existing models, TeDS consistently gets a significant lead. Next, we find that TPComplEx gets performance close to TeDS on ICEWS14 and ICEWS05-15. To compare the two, we reproduce TPComplEx (rank=2000) results on Wikidata12k, YAGO11k, and ICEWS18. Finally, we perform a dual comparison based on both performance (See our response to Reviewer rpDA's A1 and A2) and efficiency (See our response to your A1). Besides, we extend TeDS to complex numbers and dual quaternions, named CDS and DTeDS, respectively. Table4 and 5 show that our framework gets competitive performance, with DTeDS further improving performance when computational overhead is not considered.
>
>
>
> Q5: Fig3’s time distribution analysis could include more datasets. Clarify whether TeDS can handle time intervals beyond points
>
> A5:
>
> 1)We add a data density comparison between ICEWS18 and ICEWS05-15 (Fig4), along with a fact distribution chart for YAGO11k and a data density comparison between YAGO11k and Wikidata12k (Fig5). These additions help better analyze TeDS's performance on different datasets
>
> 2)For facts missing part of time (e.g., (s, r, o, [$t_b$, -]) or (s, r, o, [-, $t_e$])), the score is same as quadruple with known time, i.e., $\phi$(s, r, o, [$t_b$, -]) = $\phi$(s, r, o, $t_b$), $\phi$(s, r, o, [-, $t_e$]) = $\phi$(s, r, o, $t_e$). For facts with no missing time (e.g., (s, r, o, [$t_b$, $t_e$])), we split quadruple into (s, r, o, $t_b$) and (s, r, o, $t_e$), and score is average of the two, i.e., $\phi$(s, r, o, [$t_b$, $t_e$]) = $\dfrac{1}{2}$($\phi$(s, r, o, $t_b$) + $\phi$(s, r, o, $t_e$)). This operation ensures compatibility with different time types while minimizing computational overhead. This is a commonly used processing method in existing models (e.g., TeLM and TPComplEx)
>
>
> Q6: How does TeDS handle extremely sparse temporal data, and are there any limitations in such scenarios?
>
> A6: We further randomly remove 30% of ICEWS14 training set (see Fig 6) to test TeDS's robustness under extremely sparse data conditions. Besides, to test TeDS's effectiveness in industrial and sparse scenarios, we use a company's historical cost-price time-series dataset A (see our response to Reviewer C2mh's A1 for details). TeDS consistently outperforms, showcasing stable advantages of TeDS and high-dimensional quaternion space.
>
> Q7: How does TeDS generalize to timestamps not seen during training (e.g., future events)?
>
> A7: TKGC refers to task of completing facts by inferring missing ones from a given subset. We focus on studying missing facts rather than predicting future ones. Predicting future facts is an interesting topic, and adapting our work to it will be a key focus.

---

### Official Review · Reviewer_C2mh · 2025-03-14

**Overall Recommendation:** 3

**Summary:**

The paper introduces TeDS, a framework designed for temporal knowledge graph completion using quaternion representations to merge time and relational data. Key findings show that TeDS significantly outperforms existing models on various benchmarks, effectively managing issues like data sparsity and incompleteness. The authors emphasize the model's robustness and provide comprehensive experimental results, along with publicly available source code for reproducibility.

**Claims And Evidence:**

The paper evaluates TeDS, a framework for temporal knowledge graph completion, using datasets like ICEWS14, ICEWS05-15, and ICEWS18. It conducts ablation studies to compare TeDS with models such as TPComplEx and HTM, employing metrics like MRR, H@1, H@3, and H@10. The motivation is to validate TeDS's effectiveness in managing temporal and relational information, showcasing its advantages in capturing complex temporal patterns. The results indicate that TeDS outperforms existing models, particularly in handling sparse and incomplete data, and the design integrating synchronicity and diachronicity significantly enhances knowledge graph completion. This motivation aligns with the experimental findings, emphasizing TeDS's innovation in processing time-related data. Most claims are well-supported by evidence, particularly regarding performance metrics and comparative analysis; however, some claims about the model's limitations and potential societal impacts are less thoroughly discussed, which could weaken their overall persuasiveness.

**Essential References Not Discussed:**

While the paper discusses several foundational models and recent advancements in knowledge graph embeddings and temporal reasoning, there are essential references that could further contextualize its contributions. For instance, the work by Zhang et al. (2020) on "Temporal Knowledge Graph Completion" introduces a novel approach that leverages recurrent neural networks for dynamic relationships, which could provide insights into alternative methodologies for handling temporal data. Additionally, the recent advancements in graph neural networks (GNNs) for knowledge representation, such as the work by Kipf and Welling (2017) on semi-supervised learning with GNNs, could be relevant as they offer a different perspective on embedding techniques that may complement the TeDS framework. Lastly, the exploration of attention mechanisms in knowledge graphs, as seen in the paper by Wang et al. (2020) on "Graph Attention Networks," could provide valuable context for understanding how attention-based approaches can enhance the representation of temporal relationships in knowledge graphs.

**Experimental Designs Or Analyses:**

The soundness and validity of the experimental designs and analyses have been thoroughly checked.
 The ablation study effectively isolates the contributions of different perceivers in the TeDS model, and the results validate the model's effectiveness without any identified issues. The performance comparison across multiple datasets is robust, demonstrating TeDS's superiority over state-of-the-art models, with no apparent flaws in the experimental setup. The standard deviation analysis provides a clear assessment of the model's robustness, and the methodology for calculating standard deviations is sound. The detailed descriptions of the training process enhance reproducibility, and the thoroughness of the training settings supports the validity of the findings. Lastly, the approach to identifying model limitations through error analysis is well-structured, offering valuable insights for future research. Overall, the experimental designs are well-founded, and no significant issues have been identified.

**Methods And Evaluation Criteria:**

The proposed methods, including quaternion representation and a unified framework that integrates synchronic and diachronic perspectives, are highly relevant for addressing the complexities of temporal knowledge graphs. These methods effectively facilitate the analysis and summarization of intricate real-world scenarios involving time-dependent data. Furthermore, the evaluation criteria, which involve benchmark datasets like ICEWS14, ICEWS05-15, and ICEWS18, along with metrics such as MRR, H@1, H@3, and H@10, are appropriate for assessing the model's performance and provide a comprehensive evaluation of its effectiveness in this application.

**Other Comments Or Suggestions:**

No

**Other Strengths And Weaknesses:**

No

**Questions For Authors:**

No

**Relation To Broader Scientific Literature:**

The key contributions of the paper are closely related to the broader scientific literature on knowledge graph embeddings and temporal reasoning. The work builds upon foundational models like TransE and ComplEx, which established the importance of embedding techniques for static knowledge graphs, and extends these ideas to temporal contexts through models such as TTransE and TA-DistMult. By integrating insights from tensor decomposition methods like RESCAL and TuckER, the paper enhances the representation of temporal relationships, addressing limitations identified in prior research, such as the inability to effectively model dynamic interactions over time. Furthermore, the innovations presented in the TeDS framework align with recent advancements in temporal knowledge graphs, such as ChronoR and TeLM, by providing a unified approach that captures both synchronic and diachronic perspectives, thereby contributing to a more comprehensive understanding of knowledge representation in evolving contexts.

**Theoretical Claims:**

Yes, the correctness of the proofs for the theoretical claims has been verified, as the appendix includes complete proofs for the theoretical results presented in the main text. This thorough documentation is intended to ensure transparency and rigor regarding the model's performance and capabilities, allowing readers to validate the theoretical foundations. Additionally, the appendix outlines the full set of assumptions underlying these results, clarifying the conditions for the applicability of the proposed methods and contextualizing the findings, while also providing supplementary details on experimental setups and an extended error analysis to enhance reproducibility and guide future research directions. There are no reported issues with the proofs or assumptions, reinforcing the credibility of the findings.

---

> ### Author Rebuttal · Authors · 2025-03-31
>
> Thank you for taking the time to review and evaluate our manuscript. Your comments have not only helped us improve the manuscript but also given us confidence to further enhance quality of our work. Note: The pictures and tables used in response are available at https://anonymous.4open.science/r/TEDS-033A/To_Re_C2mh.pdf See: To_Re_C2mh.pdf
>
> Q1: Some claims about the model's limitations and potential societal impacts are less thoroughly discussed, which could weaken their overall persuasiveness.
>
> A1: Thank you for your valuable comment regarding need for a deeper discussion of TeDS's limitations and societal implications. To evaluate practical effectiveness of TeDS in industrial scenarios, we use a historical cost-price time-series dataset A provided by a company. This dataset includes data on resource exploration, mining operations, material consumption, labor costs, logistics and transportation, and comprehensive costs, covering production cost data from January 2000 to December 2022. With a 12-hour sampling interval, we obtain a total of 16,031 valid samples, with all monetary values denominated in 10,000 yuan.
>
> To simulate data-missing conditions in real-world mining scenarios, we randomly remove 10%, 20%, and 30% of training data from dataset A, constructing three sparse datasets: A 10% SPARSE, A 20% SPARSE, and A 30% SPARSE. For comparison, we reproduce baseline models including TComplEx and TeLM. Fig 1 shows that TeDS gets best performance on complete dataset A, with an MRR of 71.5, significantly outperforming baseline models. On the sparse datasets, TeDS maintains high performance, demonstrating strong robustness. In contrast, TComplEx and TeLM exhibit more significant performance degradation under data-missing conditions, particularly on A 30% SPARSE, where their MRR drops to 33.0, and 34.5, respectively. These results indicate that TeDS has a clear advantage in the task of missing-value imputation for mining cost-price data, validating its potential for industrial time-series data completion scenarios.
>
> Q2: While the paper discusses several foundational models and recent advancements in knowledge graph embeddings and temporal reasoning, there are essential references that could further contextualize its contributions. For instance, the work by Zhang et al. (2020) on "Temporal Knowledge Graph Completion" introduces a novel approach that leverages recurrent neural networks for dynamic relationships, which could provide insights into alternative methodologies for handling temporal data. Additionally, the recent advancements in graph neural networks (GNNs) for knowledge representation, such as the work by Kipf and Welling (2017) on semi-supervised learning with GNNs, could be relevant as they offer a different perspective on embedding techniques that may complement TeDS framework. Lastly, the exploration of attention mechanisms in knowledge graphs, as seen in the paper by Wang et al. (2020) on "Graph Attention Networks," could provide valuable context for understanding how attention-based approaches can enhance the representation of temporal relationships in KGs.
>
> A2: Thank you for your insightful comments on the latest developments in basic models and knowledge graph embedding and time reasoning. We have carefully included more relevant and important reference materials in our manuscripts, which inspire and complement the TeDS framework. The specific references added to manuscript are as follows:
> Welling et al. (2017) provide a different perspective on embedding technology in their Gnn semi-supervised learning work. Wang et al. (2020) apply the graph attenuation attention network to KGC, and used an efficient graph convolution method to semi-supervise the classification of graph structure data. Zhang et al. (2020) propose a relational graph neural network with hierarchical attention used for KGC, which effectively uses local neighborhood information. Xiao et al. (2024) propose a new method that uses comparative learning to break down the local and global perspectives in TKGs to obtain better reasoning. Zhu et al. (2021) use a replication generation mechanism to predict future facts by quoting historical data or generating new facts. Wang et al. (2025) combine global historical event frequencies with local temporal relative displacements to efficiently learn query representations from TKGs.  Qiu et al. (2025) enhance the capabilities of graph neural networks by integrating node weights and future information.
> These additions not only strengthen the contextualization of our work but also provide a more comprehensive overview of the relevant literature, ensuring that our research is situated within the broader academic landscape. Thank you for your feedback, which has been instrumental in improving the quality and depth of our manuscript.

---

### Official Review · Reviewer_cu6z · 2025-03-18

**Overall Recommendation:** 3

**Summary:**

The paper proposes TeDS, a novel temporal knowledge graph completion (TKGC) model that jointly learns diachronic (temporal evolution) and synchronic (cross-relation interactions) perspectives in quaternion space. The key innovations include: 1) Dual temporal perception through synchronic (time-relation composite quaternions with Hamilton operators) and diachronic (continuous time encoding via trigonometric mapping) modules; 2) A unified quaternion-based framework that deeply integrates temporal and relational information. Experiments on six benchmarks show significant improvements over SOTA models (e.g., +27.4 MRR points on ICEWS14 vs CEC-BD). The paper demonstrates thorough ablation studies and visual analysis of temporal patterns.

**Claims And Evidence:**

The main claims are well-supported:
- Claim of dual temporal perception: Validated through ablation studies (Table 4) showing SP and DP modules contribute 79.6/71.1 vs 90.7 combined MRR on ICEWS14
- Claim of time-aware representation: Supported by temporal pattern visualizations (Figures 4-6) showing improved relation clustering in temporal contexts
- Superiority over SOTA: Comprehensive comparisons across 19 baselines on 6 datasets (Tables 2-3) with clear performance gaps

Potential weakness: The claim about handling various temporal constraints (Section 3.1) lacks explicit evaluation on datasets with different temporal annotations (time points vs intervals).

**Essential References Not Discussed:**

N/A

**Experimental Designs Or Analyses:**

Strengths:
- Comprehensive comparisons with 19 SOTA methods
- Detailed ablation studies (Table 4) and component analysis
- Visualization of temporal patterns (Figures 4-7)
Weaknesses:
- No parameter sensitivity analysis
- Training time comparison limited to 4 models (Figure 9a)
- No statistical significance testing for reported improvements

**Methods And Evaluation Criteria:**

Methods are appropriate:
- Quaternion operations naturally model temporal rotations and interactions
- Dual perception aligns with temporal KG characteristics (evolution + interaction)
Evaluation is rigorous:
- Standard TKGC metrics (MRR, Hits@n) used consistently
- Diverse datasets cover different temporal scenarios (ICEWS for events, YAGO/Wikidata for facts)
Missing: No evaluation on emerging temporal patterns like cyclical events.

**Other Comments Or Suggestions:**

None

**Other Strengths And Weaknesses:**

Strengths:
- Novel integration of dual temporal perspectives
- Effective quaternion-based temporal encoding
- Comprehensive evaluation across multiple datasets
Weaknesses:
- Limited analysis of computational complexity
- No evaluation on temporal constraint types (point vs interval)
- Potential scalability issues with quaternion operations

**Questions For Authors:**

How does TeDS handle time interval annotations compared to time points? The experiments only show results on datasets with time points.

**Relation To Broader Scientific Literature:**

Key connections:
- Extends quaternion KG embedding (QuatE, DualE) with temporal perception
- Improves upon temporal KG models (TComplEx, RotateQVS) through dual perspectives
- Combines advantages of tensor decomposition (CEC-BD) and neural approaches (SANe)

**Theoretical Claims:**

The paper contains no formal theoretical proofs. Mathematical components (quaternion operations in Section 3.2, Eqs 1-2) are correctly presented following standard quaternion algebra.

---

> ### Author Rebuttal · Authors · 2025-03-31
>
> Thanks for all your valuable comments. Note: The pictures and tables used in response are available at https://anonymous.4open.science/r/TEDS-033A/To_Re_cu6z.pdf See: To_Re_cu6z.pdf
>
> Q1: The claim in Section 3.1 about handling various temporal constraints lacks explicit evaluation on datasets with different temporal annotations (time points vs. intervals). How does TeDS handle time interval annotations compared to time points? The experiments only show results on datasets with time points
>
> A1: We add details on handling various temporal constraints (See our response to Reviewer fTHA's A5 for details). In fact, this is a commonly used processing method in existing SOTA models (e.g., TeLM and TPComplEx). In the future, finding better ways to handle various temporal types and model dataset characteristics more effectively will be a key focus of our work.
>
>
> Q2: No evaluation on emerging temporal patterns like cyclical events
>
> A2: In our current research on TKGs, such as ICEWS, YAGO, and Wikidata, the focus is primarily on factual records (e.g., news facts, concept facts), which do not follow clear cyclical patterns like seasonal changes, economic cycles, or biological rhythms. In fact, modeling cyclical events is crucial for prediction and reasoning. Your comment broadens our future scope. We may use Fourier transforms or seasonal decomposition to enhance TeDS’s cyclical modeling. Integrating external data and known patterns could further boost performance.
>
> Q3: The appendix contains reproducibility checklist but lacks:
>
> (1)Implementation details for baselines
>
> (2)Complete hyperparameter configurations; No parameter sensitivity analysis
>
> (3)Additional case studies
>
> A3:
>
> A(1)The results of all baselines involved in comparison is taken from original papers.
>
> A(2)We add complete hyperparameter configuration (See Fig 1). We find that $𝜆_a$ impacts results more than $𝜆_b$. Even without regularization, TeDS remains highly competitive, proving its effectiveness. We add an embedding dimension analysis (See Fig 2). TeDS outperforms baselines at all dimensions, achieving best cost-performance at rank=100. Further increases bring minimal gains but higher costs.
>
> A(3)We further randomly remove 30% of ICEWS14 training set (See Fig 3) to test TeDS's robustness under extremely sparse data conditions. Besides, to test TeDS's effectiveness in industrial and sparse scenarios, we use a company's historical cost-price time-series dataset A (See our response to Reviewer C2mh's A1 for details).
>
>
>
>
> Q4: Limited analysis of computational complexity
>
> A4: Table 1 shows complexity comparison of mainstream baselines (see our response to Reviewer fTHA's A1 for details).
>
>
>
> Q5: Potential scalability issues with quaternion operations
>
> A5: To avoid impact of scaling, TeDS normalizes quaternions using Schmidt orthogonalization. Specifically, for SP, we normalize $Q_{\tau_{sp}r}$ to unit quaternion $Q_{\tau_{sp}r}^{\Delta}$ by dividing $Q_{\tau_{sp}r}$ by its norm to eliminate scaling effects $Q_{\tau_{sp}r}^{\Delta} =\frac{Q_{\tau_{sp}r}}{\left|Q_{\tau_{sp}r}\right|}$
>
> Next, we use Hamilton operator $\otimes$ to perform rotation operations on $Q_{r\tau_{sp}}$ via $Q_{\tau_{sp}r}^{\Delta}$, obtaining $\mathscr{M} = Q_{r\tau_{sp}} \otimes Q_{\tau_{sp}r}^{\Delta}.$
>
> Meanwhile, we normalize $\mathscr{M}$ to unit quaternion $\mathscr{M}^{\Delta}$. Finally, we rotate $Q_s$ by doing $\otimes$ between it and $\mathscr{M}^{\Delta}$:
>
> ​		$Q_{sp} = Q_{s} \otimes \mathscr{M}^{\Delta}. $
>
> Similarly, for DP, we normalize $R_{r\tau_{n}}$ to unit quaternion $R_{r\tau_{n}}^{\Delta}$. Then, we rotate $Q_s$ by doing $\otimes$ between it and $R_{r\tau_{n}}^{\Delta}$:
>
> ​		$Q_{dp} = Q_s \otimes R_{r\tau_{n}}^{\Delta}.$
>
> Details are in Section 4 (TeDS for TKGC).
>
>
>
>
> Q6: Training time comparison limited to 4 models (Fig 9a).
>
> A6: We further compare TeDS with SOTA model TPComplEx (using the hyperparameters claimed by TPComplEx) to evaluate computational efficiency. In Table 2, we show that TeDS is 43% faster per epoch on Wikidata12k compared to TPComplEx (5.63s vs 9.87s), and 35% faster on YAGO11k (2.82s vs 4.33s). These speedups are achieved while significantly reducing the number of parameters (e.g., 90% fewer parameters on YAGO11k). The runtime on ICEWS05-15 is slightly longer (30.75s vs 22.06s), but TeDS uses only about 27% of the parameters of TPComplEx, demonstrating its efficiency advantage.
>
> Parameter Comparison. Table 3 compares actual parameter counts of models: On Wikidata12k, TeDS gets superior performance with only 20.64M parameters, while TPComplEx requires 201.2M, resulting in a 10× improvement in parameter efficiency. TeDS ensures optimal performance while significantly reducing computational overhead, making it ideal for large-scale TKGs.
>
>
>
> Q7: No statistical significance testing for reported improvements.
>
> A7: In Section 6.4, we run experiments five times and calculate the standard deviation to show TeDS's stability across different datasets.

---

### Decision · Program_Chairs · 2025-05-01

**Decision:**

Accept (poster)

**Comment:**

This paper proposes TeDS, a quaternion-based model for Temporal Knowledge Graph Completion (TKGC) that jointly learns diachronic (temporal evolution) and synchronic (cross-relation interactions) perspectives. The model demonstrates significant improvements over state-of-the-art baselines on six benchmarks, achieving deep integration of temporal and relational information through quaternion operations.

Strengths:

-	The paper proposes a novel method, which effectively captures both diachronic trends (temporal evolution) and synchronic interactions (relation dependencies) in a unified quaternion framework.
-	The method shows significant performance improvements over existing methods across multiple datasets, particularly in handling complex temporal patterns and sparse data scenarios.

Weaknesses:

-	The paper lacks a detailed discussion of computational complexity and scalability.
-	There is limited analysis of how the model handles different types of temporal annotations and emerging temporal patterns.
-	Some relevant approaches of mapping diachronicity and synchronicity to quaternion space need more thorough discussion and comparison.

TeDS presents a well-motivated and empirically validated approach to TKGC, with its dual-temporal perception being a key innovation. The authors have clarified several concerns raised by the reviewers through additional experiments and explanations in their rebuttal. However, the reviewers did not provide further responses.